# Suppressed transcript diversity and immune response in COVID-19 ICU patients: a longitudinal study

Priyanka Mehta[1,2] , Partha Chattopadhyay[1,2], Ramakant Mohite[1], Ranit D'Rozario[2,3], Purbita Bandopadhyay[2,3], Jafar Sarif[2,3], Yogiraj Ray[4,5] , Dipyaman Ganguly[2,3], Rajesh Pandey[1,2]

Understanding the dynamic changes in gene expression during Acute Respiratory Distress Syndrome (ARDS) progression in post-acute infection patients is crucial for unraveling the underlying mechanisms. Study investigates the longitudinal changes in gene/transcript expression patterns in hospital-admitted severe COVID-19 patients with ARDS post-acute SARS-CoV-2 infection. Blood samples were collected at three time points and patients were stratified into severe and mild ARDS, based on their oxygenation saturation (SpO$_2$/FiO$_2$) kinetics over 7 d. Decline in transcript diversity was observed over time, particularly in patients with higher severity, indicating dysregulated transcriptional landscape. Comparing gene/transcript-level analyses highlighted a rather limited overlap. With disease progression, a transition towards an inflammatory state was evident. Strong association was found between antibody response and disease severity, characterized by decreased antibody response and activated B cell population in severe cases. Bayesian network analysis identified various factors associated with disease progression and severity, viz. humoral response, TLR signaling, inflammatory response, interferon response, and effector T cell abundance. The findings highlight dynamic gene/transcript expression changes during ARDS progression, impact on tissue oxygenation and elucidate disease pathogenesis.

## Introduction

Severe cases of COVID-19 often result in acute respiratory distress syndrome (ARDS), a life-threatening condition characterized by severe lung inflammation and compromised oxygenation, as with many other respiratory infections (1). ARDS poses significant challenges in patient management and requires a comprehensive understanding of the underlying host response at a molecular level. However, it is often underdiagnosed in intensive care settings (2). A key clinical implication of ARDS is the fluctuating oxygenation status, as indicated by the oxygen saturation (SpO$_2$/FiO$_2$) ratio (3). This can serve as a critical parameter to stratify patients and study the heterogeneity of the disease (1, 4). To unravel the intricate mechanisms driving disease progression, a thorough characterization of the host transcriptome is essential.

Transcriptomic data analysis is one of the common methods that provide a comprehensive understanding of the host response and multiple studies have explored the host immune response during COVID-19 infection (5, 6). Although this method allows for high-resolution dynamic gene expression profiling, the bulk of RNA-seq based investigations focus only on genes rather than individual/specific transcripts per gene. This is in part because of the fact that gene-level studies are thought to be more robust and produce more empirically actionable results (7, 8). However, it is important to recognize that transcript-level analysis provides a granular understanding of alternative splicing, isoform expression, and post-transcriptional modifications, which can uncover novel insights into gene regulation and biological processes (9). This is especially important as collaborative international efforts have highlighted pervasive transcription with evidence of average five transcripts/gene (10). By capturing the dynamic nature of gene expression at the transcript level, transcriptomic data provide a deeper understanding of the complexity and heterogeneity of the host response.

An important aspect of host response is its dynamic nature. Numerous studies have focused on the time-dependent variations in immune and inflammatory responses in COVID-19 (11); however, a longitudinal transcriptomic understanding on the progression of ARDS in intensive care unit (ICU)-admitted COVID-19 patients is essential to better understand the underlying molecular mechanisms driving ARDS in COVID-19 patients. By analyzing gene and transcript expression patterns, we can identify dysregulated pathways and potential therapeutic targets and the underlying mechanism driving such changes. A transcriptome analysis over time can aid in the prediction of disease severity, aiding in early identification of individuals at high risk and the decision making towards relevant therapies. Thus, in this investigation, we looked at

[1]Division of Immunology and Infectious Disease Biology, INtegrative GENomics of HOst-PathogEn (INGEN-HOPE) Laboratory, CSIR-Institute of Genomics and Integrative Biology (CSIR-IGIB), Delhi, India   [2]Academy of Scientific and Innovative Research (AcSIR), Ghaziabad, India   [3]IICB-Translational Research Unit of Excellence, CSIR-Indian Institute of Chemical Biology, Kolkata, India   [4]Infectious Disease and Beleghata General Hospital, Kolkata, India   [5]Department of Infectious Diseases, Shambhunath Pandit Hospital, Institute of Postgraduate Medical Education and Research, Kolkata, India

Correspondence: rajeshp@igib.in

the longitudinal alterations in expression patterns in severe COVID-19 patients with ARDS, admitted to the ICUs of ID and BG Hospital in Kolkata, India. Longitudinal blood samples were collected from these patients at three distinct time points after the onset of ARDS: day 1 (T1), 3–4 d later (T2), and 7 d after the first collection (T3). The collection intervals allowed us to capture gene expression dynamics during disease progression and clinical outcome.

In our investigation, we have examined both transcript and gene-level differential expression patterns. By exploring these two levels of complementary analysis, we aim to assess which analytical approach provides a comprehensive understanding of the biological processes at play delineating host immune response. The transcriptomic data are integrated with clinical and biochemical parameters to capture the complex interactions and relationships between immune cell types, cytokines, clinical parameters, and gene expression patterns. In summary, our study highlights the reduction in transcript diversity as time and severity increase, association of immunoglobulin genes with disease severity and a shift towards pro-inflammatory state overtime (Fig 1).

# Results

## Clinical demographics and patients classification

In this study, longitudinal blood samples were collected from COVID-19 patients admitted to ICU at the ID and BG Hospital in Kolkata, India (Fig S1A). During the period of hospitalization, blood was collected at three time-points at an interval of 3–4 d after admission: day 1 (T1), 4 d later (T2), and 7 d after first collection (T3) (Fig 2A). All the patients suffered from severe COVID-19, according to Indian Council of Medical Research guidelines, symptoms and showed evidence of ARDS as measured by their oxygen saturation (ratio of oxygen saturation [$SpO_2$] and fraction of inspired oxygen [$FiO_2$]) which ranged between 100–300 mmHg. To evaluate the respiratory status across the ARDS patients, the area under the curve (AUC) was computed as described earlier from the $SpO_2/FiO_2$ ratio kinetics curve over 7 d post admission (4, 12). Despite all the patients having severe COVID-19, their $SpO_2$ curve varied from 100 to 2,000 mmHg. The patients were segregated into two groups based on AUC values. Patients with an AUC ≤771.7 were categorized as Low AUC and had median $SpO_2/FiO_2$ ratios between 90–150 mmHg, whereas those with an AUC > 771.7 were designated as High AUC with median $SpO_2/FiO_2$ ratios between 100–350 mmHg (Fig 2B). Between the two groups, there was no difference in the median age of the patients (low = 59 [22–82] yr; high = 56.5 [46–84] yr). Whereas all the patients in the High AUC group recovered, three patients in the Low AUC group resulted in mortality (Table S1). In the Low AUC group, only one patient received convalescent plasma therapy (CPT), whereas in the High AUC group, six patients underwent the same treatment. In addition, a few patients in both groups also received remdesivir, ivermectin, and corticosteroids such as dexamethasone, hydrocortisone along with other drugs as part of their treatment regimen (Table 1). Type 2 diabetes and hypertension were amongst the common comorbidities observed in both the groups. Furthermore, there were no significant differences observed in the biochemical parameters between the patients in both the groups.

## Transcript expression patterns reveal suppressed transcript diversity and B-cell response during disease progression

According to the Ensembl (GRCh38) genome reference, there are ~20 k protein coding and ~17 k noncoding genes in humans which transcribe more than 150 and 50 k distinct transcripts, respectively (Fig 2C). During infection, the transcriptome dynamics has been shown to vary depending on the type of infection and the severity of the disease condition. Therefore, we examined the expression dynamics across the time points from the RNA-seq data at both the gene and transcript isoform levels to derive granular understanding of the patient transcriptome through disease progression. We begin by looking at the transcript-level expression patterns. We observed that 133 transcript isoforms varied significantly between T3 and T1, 107 transcripts down-regulated and 26 up-regulated at $P$adj < 0.05 and log$_2$FC |1| (Fig 2D) (Table S2). However, across other time point comparisons (T1 versus T2 and T2 versus T3), no significant transcripts were found. We filtered any unannotated transcripts and pseudogenes (n = 20) for further consideration. We then look at the biotypes of each transcript because various isoforms of the same gene were found to be significantly expressed. We find most of the transcripts fall into protein-coding biotypes (n = 67), whereas 15 transcripts were annotated with retained introns, 10 transcripts with nonsense mediated decay, four were lncRNAs and nine processed transcripts. We also observed the presence of several immunoglobulin heavy and light chain transcripts (IG), two for constant (C) and seven for variable regions (V) in the significant isoforms (Fig 2E). Interestingly, all the immunoglobulin transcripts are down-regulated at T3 compared with T1.

It is important to note that we observed multiple isoforms for some of the differentially expressed transcripts (*IL18R1*, *IFI27*, *SEMA4D*, *SIGLEC1*, *FKBP5*, *TTN*, *SERPINA1*, *DHRS13*, *SYNE1*, *NAIP*, *MAP2K6*, *KIAA1958*, *TLR5*, and *CDK5RAP2*) (Fig 2F). Majority of these transcripts are down-regulated at T3, and the decreased level of these gene isoforms are involved functionally in decreased B cell and T helper cell activation, type I interferon function, and increased inflammatory cytokine response mediated by TNF-α and TLR signaling pathways (13, 14, 15, 16, 17, 18, 19). Independent investigations have also found that genes related with some of these transcripts (*IGI27*, *SEMA4D*, *SIGLEC1*, and *SERPINA1*) are repressed in severe COVID-19, underlining their potential as early predictors of severity (20, 21, 22). Of the total differentially expressed transcripts, 61% of the transcripts were protein coding transcript at T3, compared with the 48% protein coding transcript at T1 (Fig S1B and C). We also observed an overall decreased transcript diversity (measured by the number of significantly expressed transcripts of the gene versus the number of nonsignificant transcripts expressed) at T3 compared with T1 (Fig 2G). As the infection progresses from T1 to T3, the findings show an offset in transcript diversity focused at transcription of more protein-coding transcripts in response to the infection.

Next, we performed gene-level differential expression analysis across time points to compare with transcript-level analysis. We found 118 genes to be differentially expressed between T3 and T1 ($P$adj < 0.05 and log$_2$FC |1.5|), with 78 being down-regulated, whereas 40 genes were up-regulated at T3 (Fig S1D) (Table S3). Overall, there was an overlap of 27 genes between gene and transcript level DE, with two genes, *KIAA1958* and *HIC1*, having

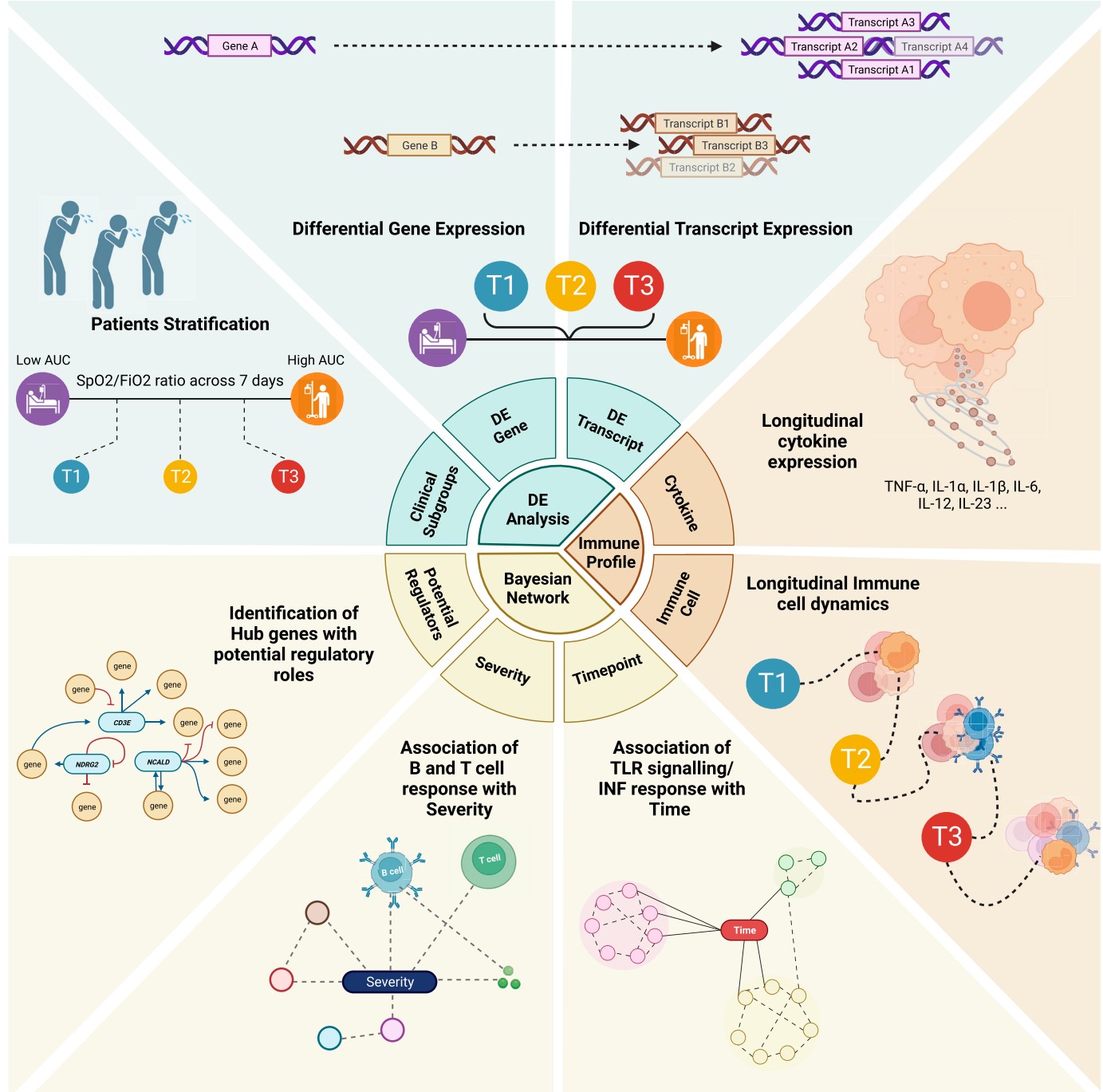

**Figure 1. Conceptual summary of the study design, illustrating the integration of multi-variable omics data for longitudinal analysis to investigate gene- and transcript-level expression dynamics over time, with a focus on understanding the severity of ICU-admitted severe COVID-19 in patients with ARDS.**

multiple significantly expressed isoforms (Figs 2H and S1E). Even though the direction of expression was comparable at the gene and transcript levels, only 20% of the differentially expressed transcripts (DTEs) were captured at the gene level. We used gene set enrichment analysis to investigate if the functional implications of all the DE genes and transcripts overlapped. We found immune response-related pathways like anti-inflammatory response, FC receptor-mediated NF-κB, MAPK and phagocytosis pathways, adaptive immune response, and BCR activation pathways to be negatively enriched at both the gene and transcript levels. However, several pathways like cell cycle, scavenging of heme from plasma, vesicle-mediated transport, cytokine and interleukin signaling pathways, and RNA pol II transcription, ESR-mediated signaling pathways were negatively enriched at only the transcript level

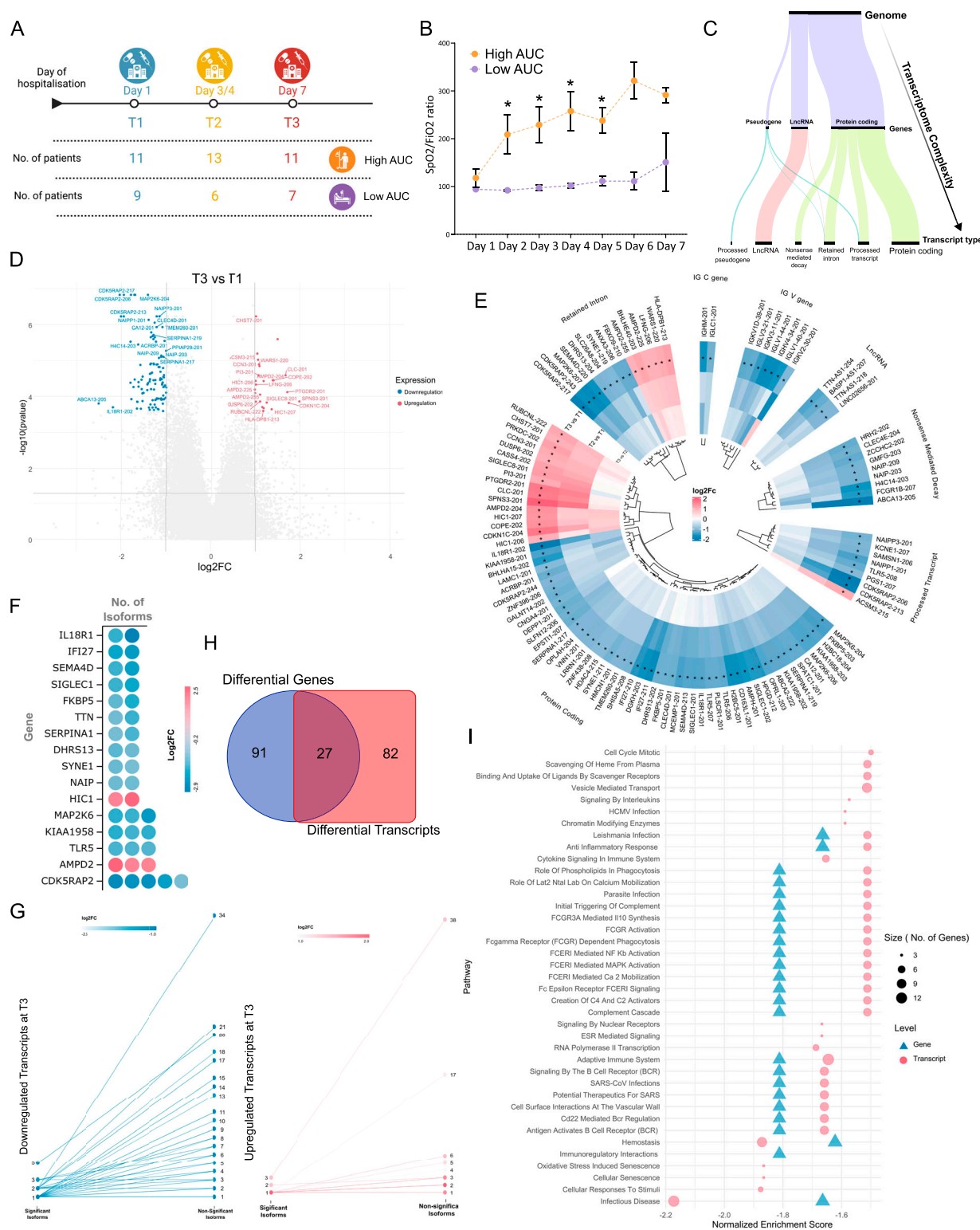

**Figure 2. Differential transcript expression between ICU-admitted COVID-19 ARDS patients across longitudinal profiling between T3 and T1.**
**(A)** Blood PBMCs were collected at three time-points–T1, T2, and T3, for patients and classified into two groups of Low AUC and High AUC. **(B)** The $SpO_2/FiO_2$ ratio curve for patients across 7 d from the first day of sample collection divides the patients into Low AUC and High AUC groups. **(C)** Global transcriptome diversity across different biotypes of genes for the human genome, representing higher transcript proportion to genes. **(D)** Volcano plot representing differentially expressed transcripts between T3 and T1. **(E)** Circular heatmap representing the $log_2$ fold change for the differentially expressed transcripts (*represents the significantly expressed $log_2$ fold changes). **(F)** The dot plot represents transcripts with multiple significantly expressed isoforms; the color corresponds to the log twofold change between T3 and T1. **(G)** The slope plot represents the number of nonsignificant isoforms expressed for each significant isoform for the up-regulated and down-regulated transcripts between T3 and T1.

(Fig 2I). This suggests that transcript level DEs highlight the alteration of cellular processes vital to maintaining homeostasis and strengthening immunological response during SARS-CoV-2 infection, in addition to the host immune response.

To validate our analysis of transcript-level expression, we conducted Leafcutter analysis comparing T1, T2, and T3 time points. We identified 133 clusters with significant differential splicing between T3 and T1. Out of these, nine clusters were found to overlap with differentially expressed transcripts (*CDK5RAP2*, *EPSTI1*, *FBXO9*, *FCGR1B*, *NAIP NAIPP2*, *SERPINA1*, *SHISA5*, and *SYNE1*). Interestingly, no transcripts were found to overlap between the T2 and T3 time point comparison. However, we did observe significant differential splicing in 30 genes between T2 and T1 (Table S4). The *CDK5RAP2* gene exhibits five significantly differentially expressed transcripts with distinct biotypes (protein-coding, retained intron, processed transcripts) (Fig 2E). Our investigation into the junctions for this transcript revealed differential splicing across three exons (Fig 3A). Specifically, junction "*a*" demonstrated an up-regulation at T3 (with a dPSI of 0.325), whereas both junctions "*b*" and "*c*" exhibited down-regulation at T3 (with dPSIs of −1.40 and −0.135, respectively) (Fig 3B). Notably, junctions "*b*" and "*c*" corresponded to the processed transcript biotype isoforms, whereas junction "*a*" corresponded to the biotype associated with protein-coding or retained intron (Fig 3C). Despite all these transcripts experiencing down-regulation at T3, we observed a higher occurrence of exon-skipping events at T3. This suggests an inclination towards the production of protein-coding transcripts at the T3 time point compared with T1.

### Transcript expression indicates dysregulated T-cell activation in severe ARDS patients

As we observed a significant difference in the $SpO_2$ between the high and low AUC patients, we performed differential transcript and gene expression analysis between high and low AUC within each time-point (Fig 4A). The patient subgroups did not differ at T1, but we observed 63 transcript isoforms to be differentially expressed at T2 (12 up-regulated and 51 down-regulated, *P*adj < 0.05 and log2FC |1|) (Fig 4B) (Table S5). Whereas at T3, 105 transcript isoforms were differentially expressed (50 up-regulated and 55 down-regulated, *P*adj < 0.05 and log2FC |1|) (Fig 4C) (Table S6). Interestingly, there was an overlap of only eight transcripts (*ABI3*, *BCL11B*, *CD5*, *CR1*, *EST1*, *PVRIG*, *SLC25A38*, and *TRBC1*) between the AUC comparisons at T2 and T3 (Fig 4D). The down-regulation of *CD5*, *BCL11B*, and *PVRIG* transcripts in the low AUC group of both T2 and T3 suggest a decreased T cell activation, TCR signaling, and an increased T cell exhaustion (Fig 4E) (23, 24, 25, 26). When the transcripts are classified by biotype, we discover that most of the transcripts are protein coding (79% DTE at T2 and 68% DTE at T3) (Fig 4F). Interestingly, we observed all the significant isoforms of T cell receptor genes to be down-regulated in low AUC patients over time, pointing towards a decreased T cell

response in the severe COVID-19 patients (27, 28, 29). Similar to our previous finding, we observed a decreased transcript diversity for the differentially expressed transcripts in the low AUC group at both T2 and T3 (*P*-value = 0.0015 at T2 and *P*-value = 0.014 at T3) (Fig 4G and H). However, the protein coding transcript isoforms was less abundant in the low AUC group compared to the High AUC at both T2 (25% protein coding transcript in low AUC, 92.15% protein transcripts coding in High AUC; *P*-value 0.001) and T3 (58% protein coding transcript in low AUC, 78.18% protein transcripts coding in High AUC; *P*-value 0.026) (Fig S2A and B). We also see an increase in lncRNA at T3 in the Low AUC group (*P*-value 0.01). We hypothesized that the decreased transcript diversity and transcription of protein coding isoforms are associated with a more severe form of the SARS-CoV-2 infection.

When we performed differentially expressed genes (DGE) analysis between the Low and High AUC groups at both T2 and T3, we observed 242 DE genes at T2 (34 up-regulated and 208 down-regulated in the Low AUC) and 363 DE genes at T3 (150 up-regulated and 213 down-regulated in the Low AUC) (*P*adj 0.05 and log2FC |1.5|) (Fig S2C and D) (Tables S7 and S8). The genes followed a similar trend in the abundance of protein coding genes in the low AUC group compared with High AUC at both T2 and T3 (*P*-value 0.001) (Fig S2E and F). Despite having little overlap with the differentially expressed transcripts at T2 and T3 (Fig 4I and J), the expression profile of the overlapping genes and transcripts that were differentially expressed were similar at both T2 and T3 (Fig 4K and L). To understand the function of the DE genes and transcripts between the high and low AUC groups at both the time points, we performed pathway enrichment analysis which highlighted distinct pathways associated with transcripts that were not associated with the DE genes otherwise (Fig 4M). For example, pathways related to translation and amino acid metabolism were positively enriched in the low AUC group at T2, suggesting a dysregulated translation and amino acid metabolism process in the severe COVID-19 patients (30). We also observed a positive enrichment of the neutrophil degranulation pathway, a hallmark of severe COVID-19 infection, in the low AUC group at T2 (31). On the other hand, the PD-1 pathway and CD28-mediated co-stimulatory pathway were negatively enriched in the low AUC group at T3, indicating a dysregulated T cell activation in the severe COVID-19 patients (32). The TCR signaling pathway was also negatively enriched at both T2 and T3 in the low AUC group, further suggesting the same. Overall, the pathway enrichment analysis highlights dysregulated metabolic function and T cell-mediated immune response in the low AUC group at T2 and T3. At the same time, it also highlights that transcript level analysis of the host response can provide deeper insights about the disease state during an infection (33). Interestingly, two pathways, adaptive immune response and IL-1 signaling pathway were negatively enriched based on transcript expression, but positively enriched based on the gene expression profile in the low AUC group. To gain a better understanding of this disparity, we investigated the cytokine and immune cell profiles.

---

Multiple genes having the same number of transcripts/expressing the same number of transcripts are clubbed together. **(H)** Venn diagram represents the overlap between differential expressed genes (in blue) and transcripts (in red). **(I)** The plot represents the significantly enriched reactome pathways for DGE (as blue triangles) and DTEs (as red dots). The size of the icon represents the number of genes involved in the pathway.

**Table 1. Clinical characteristics of the patient cohort.**

| Characteristics | High AUC (n = 14) | Low AUC (n = 9) | *P*-value |
|---|---|---|---|
| Gender (F\|M) | 2\|12 | 1\|8 | 0.82 |
| Age | 56.50 (22–82) | 59 (46–84) | 0.28 |
| Comorbidity | | | |
| Diabetes type II[a] | 5 (35.71%) | 5 (55.55%) | 0.99 |
| Hypertension[a] | 5 (35.71%) | 5 (55.55%) | 0.99 |
| Dislipidemia[a] | 1 (7.14%) | 2 (22.22%) | 0.52 |
| Hypothyroid[a] | 0 (0%) | 2 (22.22%) | — |
| COPD[a] | 0 (0%) | 2 (22.22%) | — |
| Additional therapy | | | |
| Plasma | 6 (42.85%) | 1 (11.11%) | 0.01 |
| Remdesivir[a] | 2 (14.28%) | 4 (44.44%) | 0.31 |
| Ivermectin[a] | 7 (50%) | 5 (55.55%) | 0.79 |
| Doxycycline[a] | 5 (35.71%) | 6 (66.66%) | 0.62 |
| Hydroxychloroquine[a] | 3 (21.42%) | 4 (44.44%) | 0.62 |
| Enoxaparin[a] | 9 (64.28%) | 7 (77.77%) | 0.13 |
| IV Dexamethasone[a] | 5 (35.71%) | 5 (55.55%) | 0.9 |
| Oral Dexamethasone[a] | 5 (35.71%) | 2 (22.22%) | 0.14 |
| Ambroxol[a] | 5 (35.71%) | 3 (33.33%) | 0.34 |
| Atorvastatin[a] | 5 (35.71%) | 2 (22.22%) | 0.14 |
| Biochemistry data | | | |
| Platelet count ($10^9$/liter)[a] | 233 (160–470) | 195 (127–275) | 0.36 |
| Neutrophil count ($10^9$/liter)[a] | 85 (66–91) | 77 (60–87) | 0.36 |
| White blood cell count (WBC) ($10^9$/liter)[a] | 11,150 (5,700–17,100) | 7,850 (4,100–138 k) | 0.42 |
| Lymphocyte count ($10^9$/liter)[a] | 13 (5–30) | 21 (10–35) | 0.30 |
| Red blood cell count (RBC) ($10^{12}$/liter)[a] | 4.19 (3.97–5.95) | 4.5 (3.17–5.54) | 0.90 |
| Globulin (g/dl)[a] | 2.75 (1.90–3.40) | 2.7 (2.340–3.40) | 0.89 |
| Albumin (g/dl)[a] | 4.1 (3.70–4.40) | 4.1 (3.61–4.80) | 0.97 |
| Total protein (g/dl)[a] | 6.85 (6.0–7.40) | 7.1 (6.0–7.70) | 0.97 |
| Alkaline phosphatase (ALP) (U/liter)[a] | 102 (47–163) | 84 (56–127) | 0.45 |
| Serum glutamic oxaloacetic transaminase (SGOT) (U/liter)[a] | 60 (38–84) | 58 (49–73) | 0.77 |
| Serum glutamic pyruvic transaminase (SGPT) (U/liter)[a] | 43 (22–77) | 88 (23–164) | 0.28 |
| Bilirubin (mg/dl)[a] | 0.8 (0.52–1.10) | 1.1 (0.38–1.50) | 0.77 |
| Urea (mg/dl)[a] | 35 (25–51) | 34.5 (20–46) | 0.86 |
| Creatinine (mg/dl)[a] | 1.09 (0.77–2.05) | 1.1 (0.86–1.180) | 0.92 |

Data represented as median (IQR) or n (%).
[a]Incomplete data points in either group.

## Integrated analysis reveals immune response dynamics and key associations with disease progression

Because we have differential abundance of the immune/inflammatory pathways and counter-acting enrichment patterns of two immune-related pathways at the same time, we wanted to characterize the immune/inflammatory response further (Fig 4M). We quantified the plasma cytokines of these patients across the time points. The cytokine expression across time points are available as Table S9. The B cell, T cell, and dendritic cell subsets were also quantified using specific antibodies across time points (Fig 5A). The complete cell type abundance for each individual is available as Table S10. We observed IL-18 to be significantly decreased (*P*-value = 0.029) at T3 compared with T1 (Fig 5B), whereas macrophage migration inhibitory factor (MIF) level was significantly high (*P*-value = 0.0034) at T3 compared with T1 (Fig 5C). SCGF-b, IL-16,

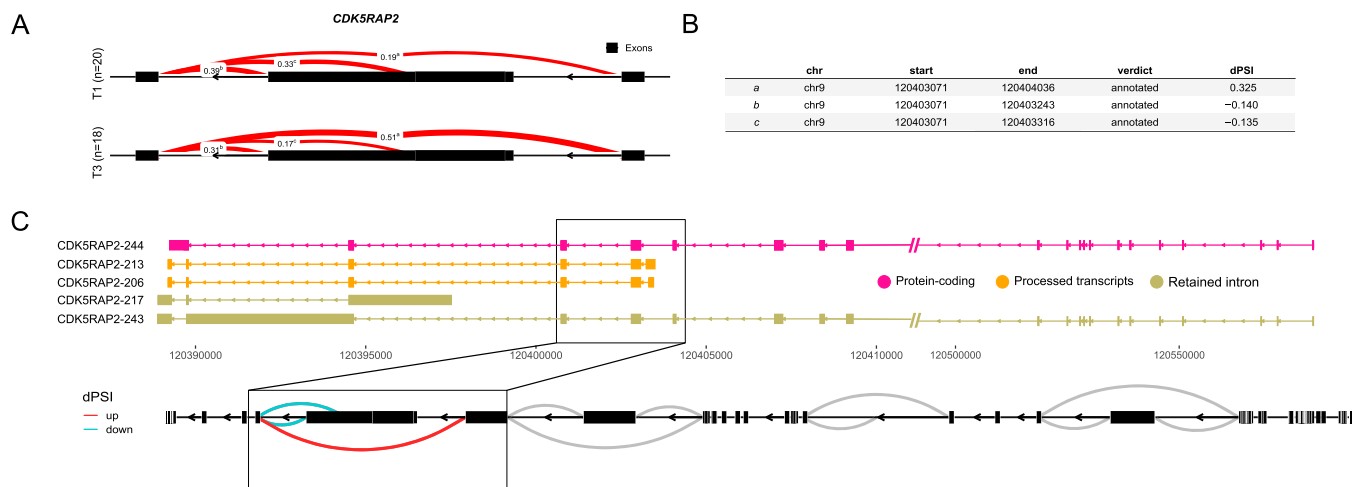

**Figure 3. Alternative splicing serves as a mechanism influencing the expression of distinct isoforms.**
**(A)** Leafcutter visualization illustrates the differential splicing of the *CDK5RAP2* gene exon junctions between the T3 and T1 time points. **(A, B)** The statistics presents junctions, indicating dPSI (Delta Percent Spliced In) values for the three exon junctions of *CDK5RAP2* as depicted in (A). **(C)** An outline of the expressed transcripts for *CDK5RAP2* includes their biotypes, with emphasis on the junction featuring reads subject to differential splicing.

and IL-6 were also differentially expressed at different time points (Fig 5D–F). The activated B cell and T helper cell populations were also decreased at T3 compared with the T1 (Fig 5G–I), suggesting a decreased B and T cell response at T3, conforming to our initial finding. Whereas the higher expression of MIF was reported to be associated with severe COVID-19 (34), down-regulation of IL-18 is possibly because of the decreased T cell function (35).

To summarize the findings related to the progression of the disease from T1 to T3, we constructed a Bayesian network model. This model incorporated the differential expression of genes and transcripts between T1 and T3, cytokine expression, cell type abundance, and clinical parameters including AUC and outcomes. We observed two prominent nodes consisting of AUC and time, suggesting that the AUC is a time-independent clinical variable, whereas the outcome is dependent on AUC (Fig 5J). It is noteworthy that we identified an association between *CNGA4*, a nucleotide-gated channel, and time. CNGA4 was found to be associated with transcripts of *MAP2K6*, *CDK5RAP2*, *TLR5*, and *CLEC4D*, and the IL-8 cytokine and CD4[+] T cell abundance. This association indicates the involvement of TLR signaling and inflammatory response, and it exhibits variability over time. Another interesting finding is the presence of a node comprising primarily T helper cells, which is associated with time. This suggests a potential association between the T helper cell population and disease progression, supporting our previous observation of decreased T helper cell function at time point (T3). We also observed a correlation between time and the pro-inflammatory cytokine growth-related oncogene-α (GRO-α). Intriguingly, specific transcripts of *SIGLEC1*, *IFI27*, *EPSTI1*, and *KIAA1958* genes were found to be associated with GRO-α, and these genes are also involved in regulating the interferon response. In addition, we noticed a cluster containing immunoglobulin genes and transcripts, particularly the variable regions of Immunoglobulin Heavy/Kappa/Lambda chains, which showed an association with the severity (AUC). Three cytokines, IL-4, IL-13, and G-CSF were

also found to be associated with the immunoglobulin genes, and are known to be involved in the immunoglobulin response cascade during an infection (36, 37, 38, 39). The Bayesian network revealed the association of specific genes and transcripts with time (TLR signaling, inflammatory response, effector T cell function, and interferon signaling) and AUC (immunoglobulin response).

### Bayesian network reveals gene/transcript clusters associated with severity

Similar to the time point-based investigation of immune response, we also investigated the cytokine and cell type abundance between high and low AUC groups at both T2 and T3 (Fig 6A). We observed an increase in IL-6 expression in the low AUC group at T2 (Fig 6B). IL-6 has already been reported to positively correlate with infectious disease severity (40, 41). Two pro-inflammatory cytokines, eotaxin and IL-8, were also significantly differentially expressed between the high and low AUC groups at T1 but not T2 or T3 (Fig 6C and D). Importantly, IL-8, up-regulated in the low AUC group at T1, may act as early biomarkers of COVID-19 disease progression and severity (42, 43).

We also observed an overall decrease in the key immune cell types in the low AUC group. For instance, CD27[+] memory B cell population decreased (*P*-value = 0.0004) in the Low AUC at T2 compared with the High AUC (Fig 6E). Similarly, the plasmacytoid dendritic cell population was also significantly decreased (*P*-value = 0.019) in the low AUC at T2 (Fig 6F). We found the CD4[+] and CD8[+] T cell populations to be significantly decreased (*P*-value = 0.002 and 0.01 respectively) in the Low AUC compared with the High AUC (Fig 6G and H). The CD4[+] T cell population was also significantly decreased (*P*-value = 0.004) in the low AUC at T3. Whereas the depleted T cell population is a hallmark of severe COVID-19 and also conforms to our previous finding (Fig 4M), only a handful of studies reported a depleted dendritic cell population

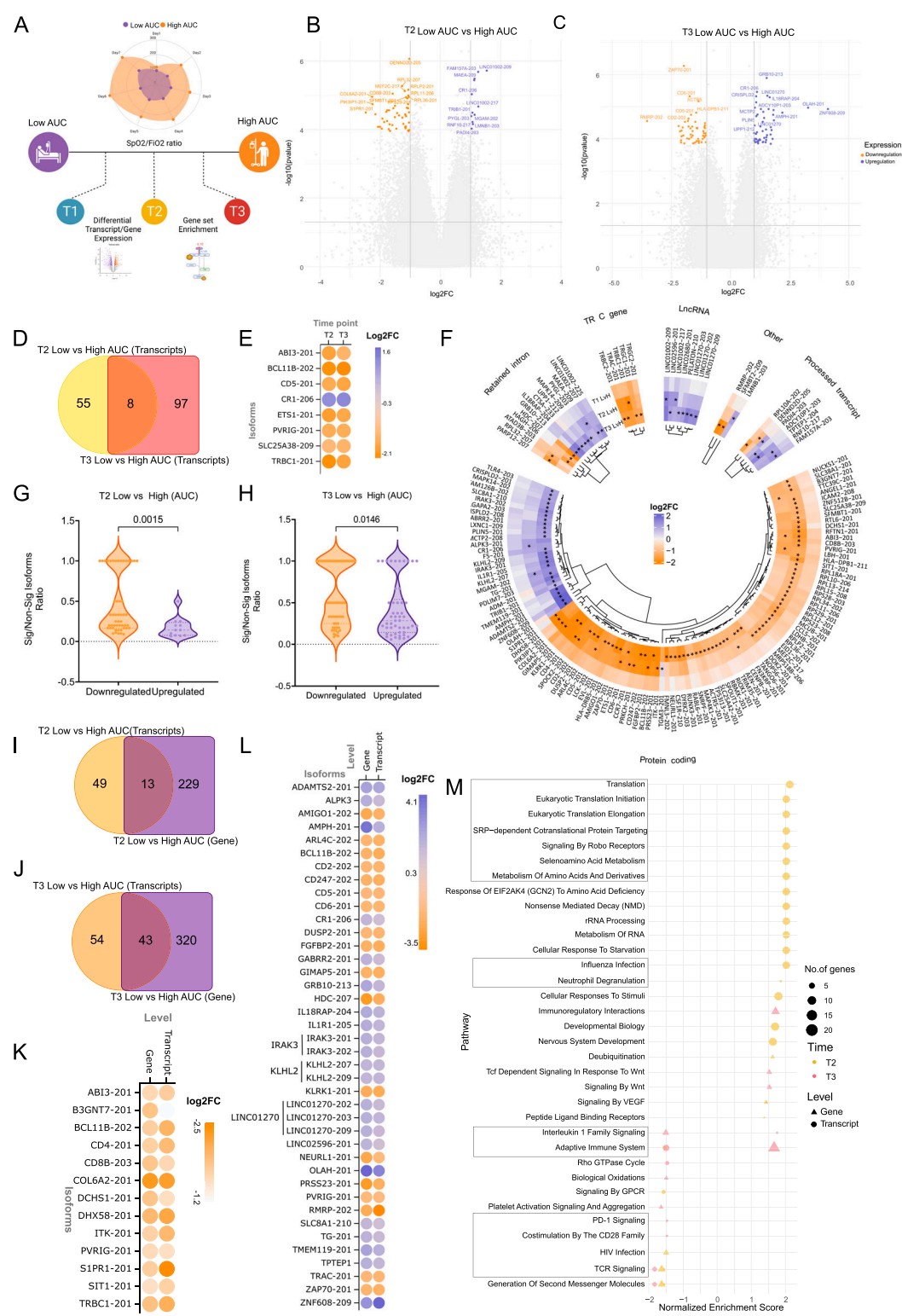

**Figure 4. Differential transcript expression between Low AUC and High AUC patients across the time points.**
**(A)** The SpO$_2$/FiO$_2$ ratio of ICU-admitted patients fluctuated over time, and each time point was investigated for DE of transcripts to elucidate variations between the two severity sub-groups. **(B, C)** The volcano plots represent the DE transcripts between Low AUC and High AUC groups at (B) T2 and (C) T3 time points. **(D)** The Venn diagram represents the overlap of DTEs between Low and High AUC groups between T2 (yellow) and T3 (red). **(E)** The dot plot represents the log$_2$fold change of the isoforms that are overlapping between the two time-points. **(F)** The circular heatmap represents the different biotypes of the isoforms expressed between Low and High AUC across time points (*represents significance in that group). **(G, H)** The violin plots represent the ratio of significant to nonsignificant isoforms between the up-regulated and down-regulated transcripts for (G) T2 Low versus High AUC group, and (H) T3 Low versus High AUC groups. **(I)** The Venn diagram represents the overlap of genes between the gene-

in severe COVID-19 and needs further investigation (44). Importantly, few studies reported a decrease in memory B cell population in vaccination breakthrough infections (45); however, there is no direct evidence of depleted B cells in severe COVID-19 in unvaccinated individuals.

Finally, we combined the differential gene and transcript expression between the two AUC groups, the cytokine profile, immune cells abundance, and the clinical parameters to build a Bayesian network to get a comprehensive picture of the differential host response between the high and low AUC patients (Fig 6I). Similar to our previous finding, we observed that the infection outcome is directly associated with AUC, but time is not associated with AUC. We observed a cluster of genes mainly regulated by *STAT4* to be associated with AUC, which plays a role in the innate immune response pathway. Another cluster controlled by the *HAPLN3*, was also associated with AUC. *HAPLN3* is involved in ERK signaling and ECM regulation, two important biological pathways altered during infection. The key transcripts present in this nodes are *CR1*, *IL1R1*, *IRAK3*, *TLR4*, and *MAPK14*, and helper CD4$^+$ T cell subtypes, and are involved in the regulation of MyD88:MAL signaling cascade, TLR signaling, and interleukin signaling. Interestingly, the AUC was also associated with *NDRG2*, a target of HIF-1a signaling and regulator of apoptosis, suggesting *NDRG2* as a possible regulator of AUC in the COVID-19 patients.

In addition, we have identified other unique clusters that play significant roles in disease-associated functions, despite their lack of association with the AUC. One cluster primarily consists of T cell receptor genes, along with other T cell–specific genes such as *CD3E*, *CD3G*, *IFI6*, *CD28*, *CD40LG*, *IFIT1*, and two subtypes of CD8$^+$ helper T cells. Another cluster is primarily regulated by the protein coding transcript of *ZAP70* and includes multiple transcripts of adaptive immune response genes like *CD2*, *CD247*, *CD5*, *CD6*, *DUSP2*, *CD8B*, and cytokines such as FGF-basic and IL-1B, along with subtypes of B cells. We also observed a third cluster consisting of transcripts of B and T cell receptor genes, and other cell type–specific genes, regulated by a protein coding transcript of *LDHB*. Notably, this cluster includes multiple protein coding transcripts of ribosomal proteins that are known to be involved in both host and viral translational regulation. These findings further support our initial discovery of dysregulated translational regulation in the low AUC patients at T2 (Fig 4M).

In summary, the Bayesian network analysis highlighted that specific signaling pathways, including interleukin signaling, MyD88:MAL signaling, TLR signaling, and innate immune response, are associated with the AUC in severe COVID-19. On the other hand, pathways related to B/T cell response, adaptive immune response, and translational regulation are found to be dysregulated, independent of their association with AUC. These findings highlight the complex interplay of biological pathways in severe COVID-19, with some pathways directly linked to disease severity and others showing dysregulation irrespective of AUC values.

## Discussion

Conventional RNA sequencing studies typically focus on gene-level expression differences, overlooking the specific expression patterns of individual transcripts. In this study, we took a comprehensive approach by analyzing both gene expression and transcript level heterogeneity in ICU-admitted severe COVID-19 patients with ARDS over time. Despite being admitted at different time points, all patients had severe COVID-19 and developed ARDS. As the disease advanced, a noticeable pattern in the SpO$_2$/FiO$_2$ ratio emerged among the patients. High AUC values implied improved oxygenation (Fig 2B).

It is worth noting that 7 out of the 23 patients (six from the High AUC group, one from the Low AUC group) in this cohort received CPT as part of the randomized clinical trial. It is important to note that the impact of the intervention was not evaluated in the current study. This is primarily because of the fact that only a small subset of individuals received the intervention, making it difficult to draw definitive and reliable conclusions regarding the long-term effects of the treatment. Besides, the randomized clinical trial involving the same patients and several other studies have reported no significant impact of CPT on the COVID-19 disease severity and outcomes (46, 47, 48, 49). Therefore, the effect of CPT on the gene and transcript expression has not been assessed in this study.

Our transcript-level analysis unveils a decline in transcript diversity over the course of 7 d in our patient cohort. This reduction is particularly significant in the higher severity group, as evidenced by Low AUC values, compared with the High AUC patients. Previous research has already indicated dysregulated transcriptional regulation and disruption of protein coding genes in COVID-19 patients (50). However, the observed decrease in transcriptome diversity in our study adds a novel aspect to our understanding of the disease and its impact on transcriptional processes. In our previous study, we also noted a similar decrease in transcript diversity among COVID-19 patients, further emphasizing the association between transcript diversity and disease severity (*P*-value 0.032) (Fig S3A) (51). This finding suggests a potential role of transcriptome dysregulation in the development of severe COVID-19 outcomes.

Interestingly, comparing gene- and transcript-level analyses showed limited overlap between them across the time points and severity groups (Figs 2H and 3I and J). This discrepancy can be attributed to factors such as the loss of transcript-specific information during gene-level quantification, variations in transcript isoform expression patterns, and technical differences in sequencing depth and sensitivity (Fig S3B–D) (33). However, gene set enrichment analysis revealed that transcript-level expression patterns effectively capture important cellular processes related to cell growth, metabolism, and maintenance of cellular functions (Figs 2I and 4M). This highlights the significance of analyzing gene expression at the transcript level, as it provides a more

level (violet) and transcript-level (orange) DE comparisons between T2 Low versus High AUC group. **(J)** The Venn diagram represents the overlap of genes between the gene-level (violet) and transcript-level (orange) DE comparison between T3 Low versus High AUC. **(K)** The dot plot represents the overlap between gene- and transcript-level comparisons of T2 Low versus High AUC with the color of the dot representing the log$_2$ fold change. **(L)** The dot plot represents the overlap between gene and transcript-level comparison of T3 Low versus High AUC with the color of the dot representing the log$_2$ fold change. **(M)** The plot represents the significantly enriched reactome pathways for DGEs (as triangles) and DTEs (as dots) at T2 (yellow) and T3 (red). The size of the icon represents the number of genes involved in the pathway.

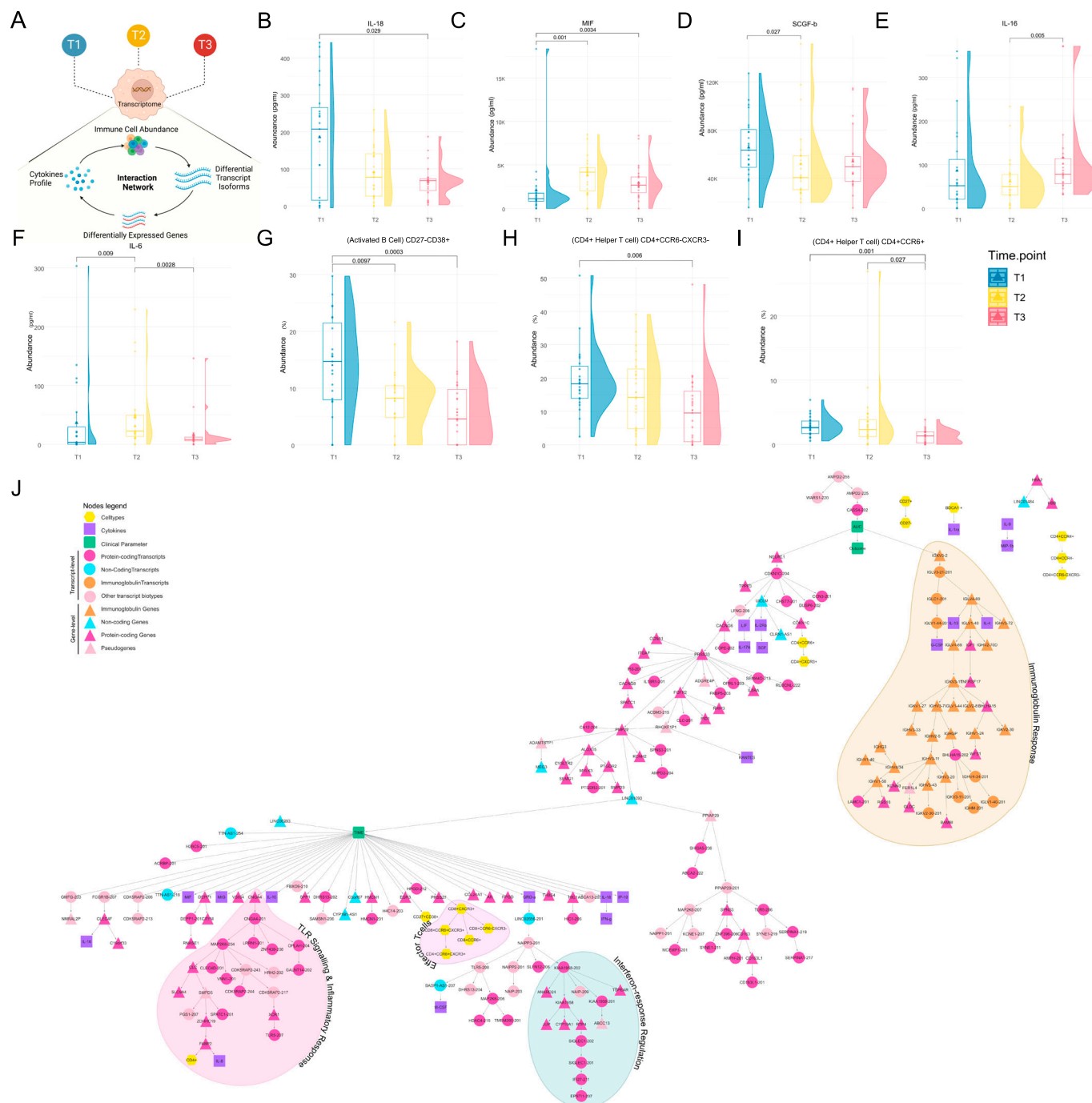

**Figure 5. Longitudinal dynamics of cytokines and cell types and association with gene/transcript expression.**
**(A)** The longitudinal dynamics of gene/transcripts and their association with immune cells and cytokines is explored using a Bayesian network. **(B, C, D, E, F)** The raincloud plots represent cytokine levels across time points (in pg/ml) T1 (in blue), T2 (in yellow), and T3 (in red) for (B) IL-18, (C) MIF, (D) SCGF-b, (E) IL-16, and (F) IL-6. **(G, H, I)** The raincloud plots represent cell types differentially abundant across time points for (G) CD27⁻CD38⁺ activated B cells, (H) CD4+CCR6-CXCR3- helper T cells (I) CD4+CCR6+ helper T cell populations. **(J)** The Bayesian network represents the connections among clusters of cytokines, cell types, differentially expressed (DE) genes/isoforms, and clinical parameters between T3 and T1. Circular nodes represent DE transcripts, whereas triangle nodes represent DE genes. The color-coded nodes represent various biotypes of genes and transcripts pink for protein-coding, blue for lncRNA, light pink for pseudogenes, orange for immunoglobulins; green nodes representing clinical parameters, purple nodes represent cytokines, and yellow nodes represent cell types.

comprehensive understanding of the underlying cellular processes during the host response compared with gene-level analysis alone (52).

As the disease advances, there is a down-regulation of inflammatory pathways, including NF-κB signaling, MAPK signaling, phagocytosis, and BCR activation at T3 (Fig 2I). This down-regulation

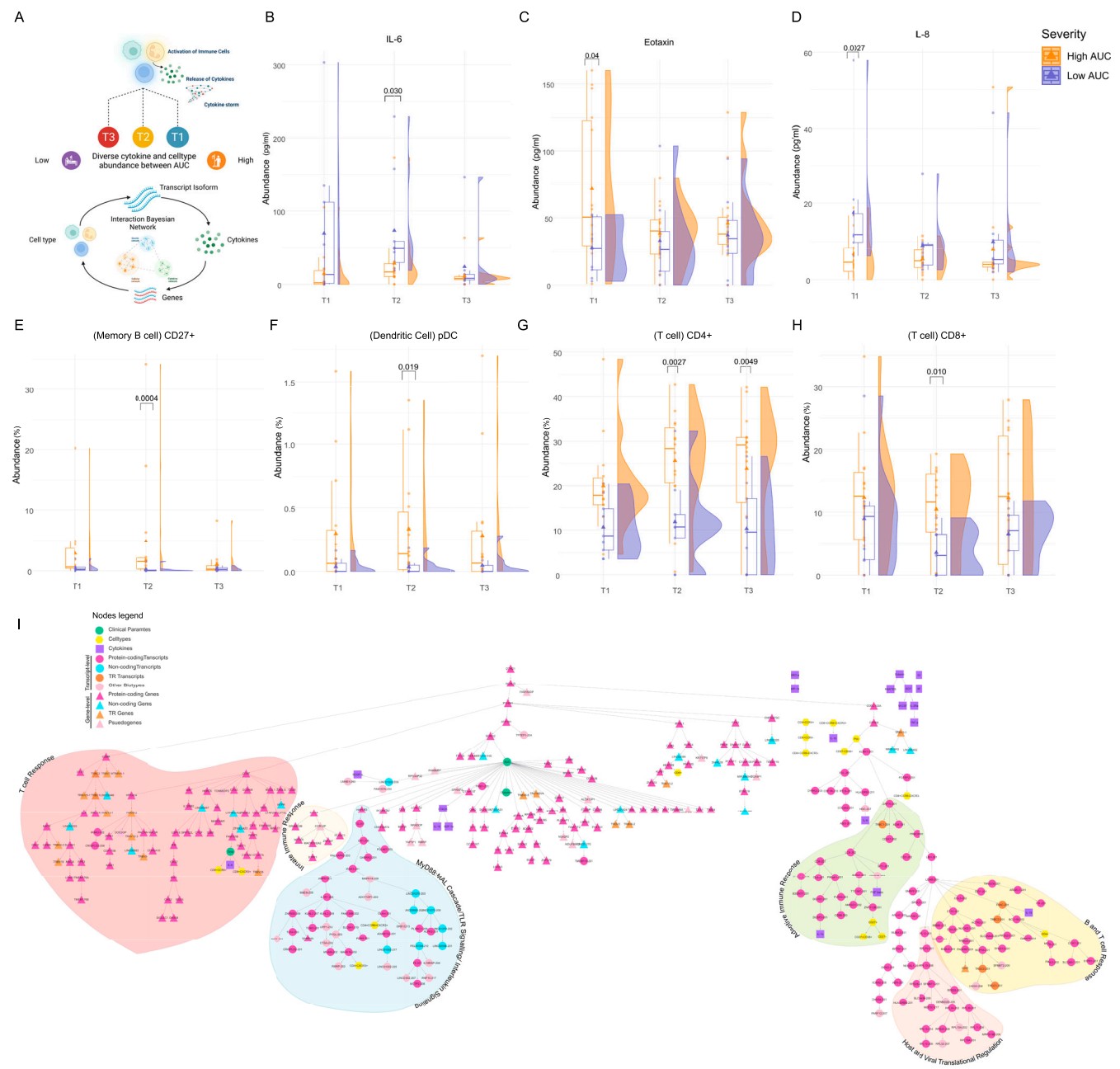

**Figure 6. Cytokines and cell type characterization in relation to severity.**
**(A)** Integrating differentially expressed genes and transcripts with differences in immune cell abundance and subsequent changes in cytokine responses to examine the interplay through time in distinct severity groups. **(B, C, D)** The raincloud plots represent cytokine levels across time points for high (in orange) and low (in purple) AUC (in pg/ml) for (B) IL-8, (C) eotaxin, and (D) IL-6. **(E, F, G, H)** The raincloud plots represent cell types differentially abundant between high (in orange) and low AUC (in purple) for (E) memory B cells CD27⁺, (F) dendritic cells pDC, (G) CD4⁺ T cell, and (H) CD8⁺ T cell. **(I)** The Bayesian network represents the connections among the clusters of cytokines, cell types, DE genes/isoforms, and clinical parameters between Low and High AUC at T2 and T3. Circular nodes represent DE transcripts, whereas triangle nodes represent DE genes. The color-coded nodes represent various biotypes of genes and transcripts pink for protein-coding, blue for lncRNA, light pink for pseudogenes, orange for T cell receptor (TR); green nodes representing clinical parameters, purple nodes represent cytokines, and yellow nodes represent cell types.

suggests a reduction in cytokines. These observations highlight the potential therapeutic value of targeting these pathways in severe COVID-19 cases to promote recovery (53, 54, 55). In our analysis of immune and inflammatory responses, we examined both plasma cytokines and immune cell abundance. Out of the 48 cytokines initially measured, we identified 36 cytokines present in plasma.

Significant changes were observed in IL-18 and MIF levels between day 1 (T1) and day 7 (T3). In addition, a dynamic expression of IL-6 was observed as the disease progressed, indicating elevated cytokine expression in the severe (low AUC) group on day 4 (42, 43, 56). In addition, we noticed a decline in activated B cells and helper T cells on day 7, aligning with our gene and transcript level findings,

indicating impaired innate and adaptive immune responses and TCR signaling (Fig 4M). Similar patterns were observed in the severe group (low AUC), suggesting that persistent viral stimulation might contribute to T cell exhaustion, leading to reduced cytokine production and impaired immune function (57, 58). Furthermore, on day 1 (T1), eotaxin and IL-8 levels were notably lower in the severe (low AUC) group. Whereas IL-8 has been linked to severe COVID-19 in some studies, the association between eotaxin expression and disease severity remains unclear (42, 43, 56). Our findings underscore the importance of gene/transcript-based analysis in gaining a better immunological understanding, surpassing the limitations of plasma cytokine quantification, which is more commonly used in clinical and research settings (59).

The Bayesian network analysis reinforced our findings on differential expression, highlighting the association between disease progression (time) and factors such as TLR signaling, inflammatory response, interferon response, and effector T cell abundance. Moreover, a group of cytokines (IL-16, MIF, monokine induced by IFN-γ, IL-10, GRO-a, IL-18, IFN-g, IP-10) demonstrated close connections to disease progression, although only IL-18 exhibited significant differences between T3 and T1 in the conventional analysis. This emphasizes the need for further investigation to fully understand the role of these cytokines in disease progression. In addition, the analysis revealed new findings, including the independence of disease severity (AUC) on time and the dependence of disease outcome on the severity (AUC) (Figs 5J and 6I). This suggests that measuring the $SpO_2$ curve (AUC) can serve as a valuable predictor of disease outcome, especially for COVID-19 patients with ARDS (60). Notably, the association between immunoglobulin response and disease severity/AUC was stronger than the association with time. Contrary to expectations, there was a decrease in immunoglobulin response and activated B cell population at T3 (Figs 2E and 5G). This could potentially be attributed to the administration of corticosteroids (dexamethasone and hydrocortisone) to the ARDS patients as part of their treatment regimen (Tables 1 and S1). Nearly 75% of the patients in both the high and low AUC groups were administered corticosteroid treatment. Studies have shown that corticosteroids modulate release of immunoglobulins, by affecting the earlier stages of B cell proliferation (61, 62). Therefore, the observed decrease in immunoglobulin response and activated B cell population at T3 may be attributed to the immunosuppressive effects of corticosteroids administered to the patients, as these drugs can directly impact B cell function and the overall immune response (63, 64).

The Bayesian network model identified clusters of genes associated with the B and T cell responses to be co-expressed in the time-dependent AUC comparison group (Fig 5I). Interestingly, both the B and T cell response pathways were associated with a calcium-binding protein family gene, NCALD. Although this gene is known to encode a neuronal calcium sensor, limited evidence shows its expression in immune cells including the T cell subtypes and the NK cells (65, 66). An interferon inducible gene, PYHIN was found to be directly associated with T cell-specific gene (67), CD3E, which in turn, was associated with a series of genes including MAF, IL12R, CD3G, CD28, CD40LG, IFI6, OAS2, cytokines such as IL6, and CD8+ T cells, regulating a cascade of T cell mediated immune and inflammatory response (68, 69, 70, 71, 72, 73). On the other hand, LDHB was identified as a possible regulator of B and T cell response (74). It was

significantly associated with T cell receptor genes TRAC and TRBC which control the antigen sensing by T cell, CCL5, which regulates the proliferation and IgM secretion by the B cell, and ETS1, which regulates the proliferation and differentiation of both B and T cells. The network suggests NCALD, PYHIN, and LDHB as possible master regulators of the B and T cell responses. However, the existing literature provides only limited knowledge on their role in regulating B and T cell response and further studies in this direction is warranted, either in different cohorts of the same disease or other infectious disease.

Furthermore, the network analysis revealed additional insights. It identified specific genes and transcripts, such as CNGA4-201 as regulators of TLR signaling and inflammatory response, KIAA1958-202 and GRO-α interferon response, providing novel associations that require further exploration (Fig 6I). The analysis also indicated that the observed decrease in T cell response and TCR signaling in severe COVID-19 may not be directly influenced by the AUC, but rather by factors such as interleukin response, MyD88:MAL signaling, TLR signaling, and innate immune response, which are regulated by specific genes and transcripts. In addition, the adaptive immune response, B/T cell response, and translational regulation pathways were found to be independent of the AUC and regulated by other genes and transcripts. The direct role of the NDRG2 gene in immune response remains unclear, but previous studies have suggested its involvement in NF-B inhibition and its potential relevance to severe COVID-19 (Fig S3E) (75). Further research is needed to fully understand the contribution of NDRG2 and other identified genes and transcripts in modulating the immune response in COVID-19.

The patient stratification method in this study, based on the $SpO_2$/$FiO_2$ ratio sheds light on the transcriptomic distinctions among COVID-19 ARDS patients with varying levels of $SpO_2$. However, clinical utility of such a stratification method is not validated in an external cohort, presenting a potential limitation in the current study. Further investigations with validation cohorts are warranted to potentially overcome/strengthen the classification method used.

In conclusion, our study highlights the importance of transcript-level analysis in capturing specific expression patterns and the association of reduced transcript diversity with disease severity (Fig 7). Gene and transcript-based investigations of the immune response provide valuable immunological insights, surpassing the limitations of plasma cytokine quantification. The Bayesian network analysis served to identify potential regulators of immune response and revealed AUC (indicative of disease severity) as a predictor of COVID-19 outcome in ARDS patients. Finally, the comprehensive analysis of gene and transcript expression in severe COVID-19 patients provides valuable insights into disease pathogenesis and offers potential for improved clinical treatment strategies in the ICU.

# Materials and Methods

### Sample collection and classification

The sample was collected from RT–PCR positive COVID-19 patients admitted to the Infectious Diseases & Beleghata General Hospital,

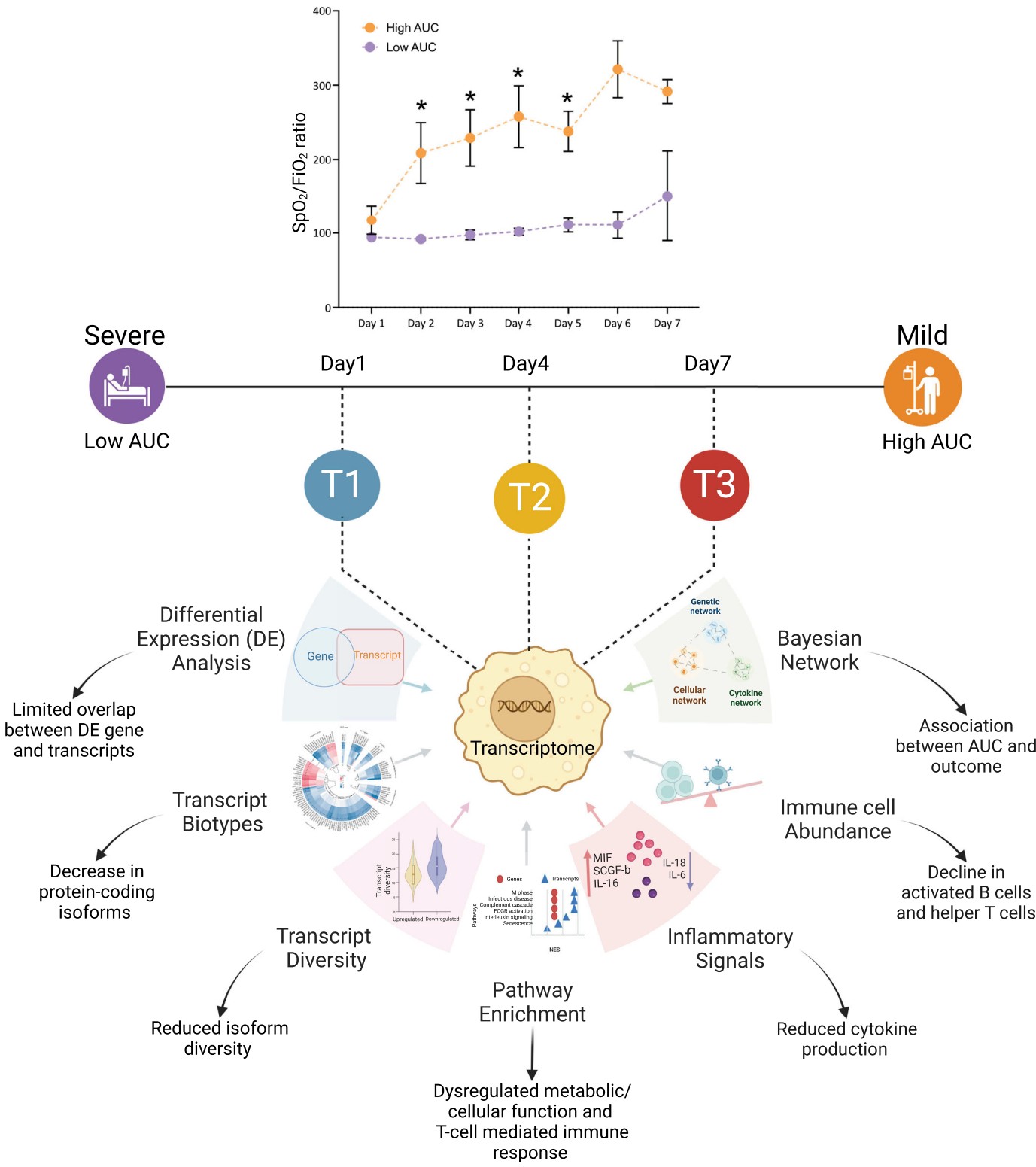

**Figure 7.  Threading together the comprehensive gene- and transcript-level inferences in ICU-admitted COVID-19 patients at three time points.**
The dynamic longitudinal analysis emphasized the significance of transcript-level inferences in complementing our understanding of the disease progression and the interplay between gene expression, cytokine levels, and immune cell activity.

Kolkata, India, with mild to moderate ARDS, recruited during a previously published clinical trial on CPT (Clinical Trial Registry of India No. CTRI/2020/05/025209) between May 31, 2020, to October 12, 2020 (4, 48). The enrolled patients were aged ≥ 18, had severe COVID-19 as per the Indian Council of Medical Research guidelines (fever or suspected respiratory infection, plus one of the following; respiratory rate >30 breaths/min, severe respiratory distress, SpO2 <90% at room air) with either mild ARDS ($SpO_2/FiO_2$ ratio between 200–300) or moderate ARDS ($SpO_2/FiO_2$ ratio between 100–200), without requiring mechanical ventilation and were within 5–10 d from initial onset of symptoms. Pregnant and breastfeeding mothers, and individuals predisposed to clinical conditions were excluded. The trial recruited 40 individuals in each arm: one which received only standard-of-care therapy as per the advisory in 2020 and CPT which received additional two consecutive doses of ABO-matched 200 ml convalescent plasma on two consecutive days, the first transfusion being on the day of enrolment, in addition to standard of care (48) (Fig S1A). At least 5 ml of blood was collected in BD Vacutainer EDTA tubes at three time points, viz. on the day of enrolment (T1), on the third or fourth day after enrollment (T2) and 7 d post enrollment (T3). Furthermore, plasma was isolated by centrifuging the blood at 2,000$g$ for 10 min at 4°C. The study was designed in accordance with the Declaration of Helsinki and was approved by the Institutional Ethics Committee of CSIR-Institute of Genomics and Integrative Biology (CSIR-IGIB), Delhi, India (Ref No: CSIR-IGIB/IHEC/2020 21/01), CSIR-Indian Institute of Chemical Biology, Kolkata, India (IICB/IRB/2020/3P), and Infectious Diseases and Beleghata General Hospital (ID and BG Hospital), Kolkata, India (IDBGH/Ethics/2429). In addition, records of the medications administered during the COVID-19 treatment for each patient were curated (Table S1). This encompassed a range of drugs, including remdesivir, and corticosteroids like dexamethasone and hydrocortisone. However, IL-6 receptor blockers and baricitinib, which are recommended in WHO guidelines, were not administered because of unavailability during the study period (76).

## Evaluation of $SpO_2$ in blood

To assess blood oxygenation levels over a span of 7 d after enrollment, we calculated the AUC for the ratio of $SpO_2$ in capillary blood to the $FiO_2$, denoted as the S/F ratio or SFR. This metric, referred to as SFR7dAUC, provided a comprehensive view of blood oxygenation trends during this 7-d period. In instances where data were not recorded on either the first or seventh day (in the event of patient survival), we imputed values from the nearest available day, giving precedence to the lower value when two such values were accessible. The resulting SFR7dAUC values were computed for each patient and used for correlational analyses with other parameters. In case of missing data between the first and seventh days (the mortality patients), the closest available value was used and replicated when calculating the SFR ratio 7-d kinetics curve. This metric was demonstrated as an effective predictor of disease severity in our cohort, with median value exhibiting robust stratification capabilities as detailed in the prior study by (4). Thus, based on the AUC values, patients were categorized into two groups. Patients with an AUC ≤ 771.7 were categorized as Low AUC and had median $SpO_2/FiO_2$ ratios between 90–150 mmHg, whereas those

with an AUC > 771.7 were designated as High AUC with median $SpO_2/FiO_2$ ratios between 100–350 mmHg representing distinct levels of oxygenation as surrogate for disease severity.

All individuals in the High AUC group made a full recovery, whereas three out of the nine patients in the low AUC group fatal outcomes to the SARS-CoV-2 infection (Table S1). Importantly, despite all patients having severe COVID-19, the $SpO_2$ curves exhibited considerable variability, ranging from 100 to 2,000. Notably, the low AUC group demonstrated poorer outcomes, with a higher mortality rate compared with the High AUC.

## Library preparation and sequencing

We used Illumina TruSeq Stranded Total RNA Library Prep Gold (cat. no 20020599) to prepare the sequencing libraries from 250 ng of total RNA extracted from the PBMC collected from patients at the three time points, as per the manufacturer's reference guide (1000000040499 v00). In summary, cytoplasmic and mitochondrial rRNA have been depleted using target-specific biotinylated oligos and Ribo-Zero rRNA removal beads, and pure RNA was fragmented using a divalent cation at a high temperature. Random primers and SuperScript IV reverse transcriptase were used to create the first strand cDNA from the fragmented RNA. After RNaseH-mediated degradation of the RNA strand from the previous phase, DNA polymerase 1 was used to synthesize the second strand of cDNA. Adenylation of the double-stranded cDNA's 3′ blunt end was done before indexing and amplification. AMPure XP bead with sample to bead ratio of 1:1 (A63881; Beckman Coulter) was used for cleaning the final library. The library quality was checked using Agilent 2100 bioanalyzer followed by sequencing on NovaSeq 6000 using NovaSeq S2 v1.5 reagents with a 2 × 101 read length and 450 pM loading concentration.

## Differential gene and transcript expression analysis

The raw sequencing reads were quality checked using FastQC and trimmed with Trimmomatic v.0.39 to remove low-quality bases (77). The reads were then reassessed with FastQC to confirm quality improvements (78). The filtered reads were then quantified against the human reference genome (GRCh38.106 primary assembly from Ensembl) using Salmon (v.1.8.0) (79). For differential gene expression analysis, the Salmon-quantified reads were analyzed using the DESeq2 package (80). For differential transcript expression analysis–numGibbsSamples parameter with 20 inferential replicates was used in Salmon to generate bootstrap abundance estimates for each sample using posterior Gibbs sampling to estimate the variability in transcript abundance. The quantification files generated by Salmon were imported to the R environment using the tximport for differential transcript-level expression analysis of RNA-seq using inferential replicate counts with Swish method in Bioconductor package fishpond (v.2.0.1) (81, 82). Differential transcript and gene-level analysis was performed for different time points (T3 versus T1, T2 versus T3 and T2 versus T1) and within the time point between different AUC groups of different severity (T3 Low versus High AUC, T2 Low versus High and T1 Low versus High). The Benjamini–Hochberg correction was used to correct for multiple comparisons (with an FDR cut-off <0.05). DGEs with $P$adj value < 0.05

and log$_2$foldchange (FC) |1.5| were considered as significant, whereas for DTE *P*adj value < 0.05 and log$_2$FC|1| was considered significant. A lower log fold change threshold for transcript-level analysis was selected to ensure capturing of more subtle differences in expression of individual transcripts, especially for detecting isoform-level changes.

## Plasma cytokine quantification

The plasma samples were obtained from the peripheral blood of the patients and collected in EDTA vials. Cytokine levels (measured in pg/ml) were determined using the Bio-Plex Pro Human Cytokine Screening Panel, 48-Plex Assay from Bio-Rad. This assay allows for the quantification of 48 different cytokines, including FGF basic, eotaxin, G-CSF, GM-CSF, IFN-γ, IL-1β, IL-1ra, IL-1α, IL-2Rα, IL-3, IL-12 (p40), IL-16, IL-2, IL-4, IL-5, IL-6, IL-7, IL-8, IL-9, GRO-α, hepatocyte growth factor, IFN-α2, leukemia inhibitory factor, monocyte chemo-attractant protein-3 (MCP-3), IL-10, IL-12 (p70), IL-13, IL-15, IL-17A, IL-18, IFN-γ–inducible protein-10 (IP-10), MCP-1, monokine induced by IFN-γ, NGF-β, stem cell factor, stem cell growth factor-β, stromal cell-derived factor-1α, macrophage-inflammatory protein-1α, macrophage-inflammatory protein-1β, PDGF-BB, regulated upon activation normal T-cell expressed and secreted, tumor necrosis factor-α, tumor necrosis factor-β, VEGF, cutaneous T cell-attracting chemokine, MIF, TNF-related apoptosis-inducing ligand, and M-CSF. To perform the assay, the patient plasma samples were diluted 1:3 in sample diluent. The assay was carried out following the instructions provided by the manufacturer. The Bio-Plex 200 System from Bio-Rad was used to run and analyze the plate.

## Cell type quantification

After plasma collection, the whole blood sample was treated with RBC lysis buffer. The leukocyte pellet thus obtained was subsequently fixed using 1% paraformaldehyde and stained using fluorochrome tagged antibodies (BD Biosciences) to analyze the different immune cell subsets based on surface marker expression. The stained samples were then acquired in a FACSAria III flow cytometer. Further analysis was performed on FlowJo Software to obtain the cell type quantification data.

## Bayesian network analysis

The data from RNA expression for both differentially expressed transcripts/genes, cytokines, and cell type abundance were integrated with AUC, outcome, and time of sampling. The integrative modeling analysis was conducted using the wiseR package, which uses end-to-end Bayesian network learning and inference techniques (83). To facilitate biological interpretation, all continuous variables within the integrated dataset were discretized into three quartile ranges: low, middle, and high. A discrete Bayesian network was then learned from the data using hill climbing optimization to identify the directed acyclic graph that represents the structural dependencies between the variables. Subsequently, the structure of the network was parameterized using Monte Carlo Markov Chain approximate inference method to determine the conditional probability distributions.

## Gene set enrichment analysis

Function enrichment of DGEs and DTEs was performed using the fgsea R package against the reactome database using c2.cp.reactome.v2023.1.Hs.symbols.gmt (84 *Preprint*). Pathways with statistically significant *P*-value cutoff <0.05 and with at least three genes were considered. The pathways were plotted using the ggplot2 R package against the combined score and number of genes involved in the pathways. Redundant pathways were excluded (85).

## Differential splicing analysis

The initial step involved aligning the pre-processed fastq files against the reference human genome (GRCh38.106) using the splice-aware aligner STAR (v2.7.11) (86). Subsequently, the resulting bam files were transformed into junction files using regtools (87 *Preprint*). After this, intron clustering and analysis of differential splicing were conducted using Leafcutter (v.0.2.9), with allowed intron sizes set between 50–500 kb (88). The Ensembl human reference (GRCh38.106 primary assembly from Ensembl) was employed for annotation. The groups for differential splicing analysis were structured in the same manner as those used for the comparison in the differential gene/transcript level analysis. The outcomes were then visualized using Leafviz, and a report was generated using RStudio (https://github.com/jackhump/leafviz) (Table S4).

## Statistical analysis and data visualization

The immune cell type and cytokine expression levels between all the time points and AUC groups were compared using the non-parametric two-way mixed-effect ANOVA model with Tukey's correction to account for unbalanced/missing data and account for individual-level variability in the dataset (89). Wherever appropriate, we compared the differences between data points using the two-tailed Mann–Whitney *U* test, and Chi-square testing. The statistical tests were performed using a licensed version of GraphPad Prism. The ggbio (v.1.44.1) (90), GenomicFeatures (v.1.48.3) (91), ggplot2 (v.3.3.3) (85), EnhancedVolcano (v.1.14.0) (92) Gviz (v.1.40.1) (93) R packages were used for data visualization. The *P*-value < 0.05 was considered statistically significant.

# Data Availability

The datasets presented in this study can be found online at the NCBI-SRA under the BioProject accession numbers PRJNA816679.

## Ethics statement

The study was designed in accordance with the Declaration of Helsinki and was approved by the Institutional Ethics Committee of CSIR-Institute of Genomics and Integrative Biology, Delhi, India (Ref No: CSIR-IGIB/IHEC/2020-21/01), CSIR-Indian Institute of Chemical Biology, Kolkata, India (IICB/IRB/2020/3P), and Infectious Diseases and Beleghata General Hospital (ID and BG Hospital), Kolkata, India

(IDBGH/Ethics/2429). The patients/participants provided their written informed consent before participation in this study.

## Supplementary Information

## Acknowledgements

The authors duly acknowledge all the COVID-19 patients who participated in the study. Authors acknowledge the help and support from Dr. Aradhita Baral and Dr Bharti Kumari towards facilitation as research manager and coordination with the funders. Authors acknowledge the support of Anil Kumar and Nisha Rawat towards COVID-19 sample transport and sample management. P Chattopadhyay acknowledges the CSIR for his Research Fellowship. This research was funded by Bill and Melinda Gates Foundation, grant numbers INV-033578 to R Pandey and MLP-129 from Council of Scientific and Industrial Research, India, to D Ganguly.

### Author Contributions

P Mehta: data curation, software, formal analysis, investigation, visualization, methodology, and writing—original draft.

P Chattopadhyay: data curation, investigation, writing—original draft and sequencing.

R Mohite: investigation and writing—review and editing.

R D'Rozario: resources and data curation.

P Bandopadhyay: resources and data curation.

J Sarif: resources and data curation.

Y Ray: resources and data curation.

D Ganguly: resources, data curation, supervision, and writing—review and editing.

R Pandey: conceptualization, supervision, funding acquisition, investigation, visualization, project administration, and writing—review and editing.

### Conflict of Interest Statement

The authors declare that they have no conflict of interest.

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
