## [Reviewer comments · Life Science Alliance]

Life Science Alliance

Suppressed transcript diversity and immune response in COVID-19 ICU patients: A longitudinal study

Priyanka Mehta, Partha Chattopadhyay, Ramakant Mohite, Ranit D'Rozario, Purbita Bandopadhyay, Jafar Sarif, Yogiraj Ray, Dipyaman Ganguly, and Rajesh Pandey

DOI: <https://doi.org/10.26508/lsa.202302305>

Corresponding author(s): *Rajesh Pandey, Institute of Genomics and Integrative Biology*

Review Timeline:

Submission Date:	2023-08-03
Editorial Decision:	2023-09-14
Revision Received:	2023-09-27
Editorial Decision:	2023-10-16
Revision Received:	2023-10-19
Editorial Decision:	2023-10-20
Revision Received:	2023-10-21
Accepted:	2023-10-23

Scientific Editor: *Eric Sawey, PhD*

Transaction Report:

September 14, 2023

Re: Life Science Alliance manuscript #LSA-2023-02305-T

Dr. RAJESH PANDEY
CSIR INSTITUTE OF GENOMICS AND INTEGRATIVE BIOLOGY
Mall Road, Delhi
Delhi 110007
India

Dear Dr. Pandey,

Thank you for submitting your manuscript entitled "Lost in transcription? Longitudinal Expression Dynamics in ICU-admitted COVID-19 patients reveal suppressed transcript diversity and immune response in severe ARDS patients" to Life Science Alliance. The manuscript was assessed by expert reviewers, whose comments are appended to this letter. We invite you to submit a revised manuscript addressing the Reviewer comments.

Thank you for this interesting contribution to Life Science Alliance. We are looking forward to receiving your revised manuscript.

Sincerely,

B. MANUSCRIPT ORGANIZATION AND FORMATTING:

Reviewer #1 (Comments to the Authors (Required)):

This manuscript by Mehta et al. investigated the longitudinal changes in gene expression patterns in hospital admitted severe COVID-19 patients with ARDS post-acute SARS-CoV-2 infection, followed in parallel with disease severity taking tissue oxygenation as the parametric surrogate. The authors applied combination of complex and dynamic variables during SARS-CoV-2 infected patients who were admitted in ICU with known clinical outcomes. Important variables included RNAseq-based longitudinal profile of ICU admitted patients to capture host immune responses, transcript isoforms' differential expression at multiple time-points, and usage of Bayesian network to identify new nodes underlying the COVID-19 disease. Their finding can strengthen a critical gap of COVID-19 disease wherein challenging integrative approach has been undertaken. Overall, this is an interesting study.

Here are specific comments for minor revision:

- 1) It would be helpful for the readers to have a conceptual figure/study summary at the start, highlighting the investigation strategy and threading of multi-variables inclusive of clinical data, cytokine profiling, cell types and importantly Genomics strategy for functional role of transcript isoforms.
- 2) One of the primary mechanisms of transcript isoforms generation is through alternative splicing. Authors should consider highlighting any/few genes with respect to alternative splicing using tool like LeafCutter, if possible, that would help strengthen the possible mechanism.
- 3) Bayesian network is an interesting component of the study. Authors should highlight any specific B or T cell mediated immune response from that analysis. These possibilities can be discussed in the discussion section, if not in results.
- 4) Thorough proofreading for grammatical errors and appropriate use of caps and small case in the text are needed. Text font size in some of the figures are difficult to read.

Reviewer #2 (Comments to the Authors (Required)):

In this study the authors sought to "investigate the longitudinal changes in gene expression patterns in hospital admitted severe COVID-19 patients with ARDS post-acute SARS-CoV-2 infection, followed in parallel with disease severity taking tissue oxygenation as the parametric surrogate". A posthoc analysis of a randomized controlled trial was performed in other to assess this study objective. Unfortunately, the study does not adequately contextualize nor assess the fact that patients received plasma convalescent therapy and its potential role on gene expression patterns, which is one of the main limitations of this study. Additionally, participants were studied according to having a "low" or "high" SpO₂/FiO₂ AUC, which is a major drawback since it has limited relevance and potential to be applied as this classification scheme was developed by the authors without prior rigorous development and validation (refer to TRIPOD statement for the limitations and dangers of such an approach <https://doi.org/10.7326/m14-0698>). Thus, such classification in low and high is at a very high risk of being meaningless since not even an internal validation procedure was performed (again, refer to TRIPOD). A third major limitation of this study is that the dependency of data (repeated measurements) was not considered in statistical analyses as the groups at different time points were seemingly incorrectly assumed to be independent.

1. There is an associated publication for the study CTRI/2020/05/025209 (<https://doi.org/10.1038/s41467-022-28064-7>), however, it looks like the authors have not included this reference describing the results of the trial. In the original trial, 80 participants were randomized into the standard of care arm (n=40) and convalescent plasma therapy (CPT) arm (n=40). Nonetheless, in this study the authors only report approximately 20 patients for time 1 (day 1); 19 patients, for time 2 (day 3-4), and 18 patients, for time 3 (day 7). Please explain why the remaining patients were not included in the study.
2. This manuscript needs to explicitly mention in the title, abstract and main manuscript that it is a post-hoc study of a randomized controlled trial.
3. Unfortunately, not giving sufficient detail on how the patients were recruited into this study fails to convey the main objective

of the recruitment strategy which was to evaluate efficacy and safety of plasma convalescent therapy. This is critical since randomization was performed to randomly distribute confounders among study groups and it is likely possible that the effect of randomization has been neglected by creating new study groups by classifying patients according to SpO₂/FiO₂ AUC. Most importantly, the authors do not give details on which patients in this study received the intervention and who did not. Please provide a flow diagram of patients who remained in this study according to treatment group (SOC vs PCT).

4. What is the effect of plasma convalescent therapy on expression dynamics? Why was this not assessed in this study? I cannot understand why the fact that the patients received this intervention was completely omitted. Substantial explanation, justification and discussion on this should be incorporated into the manuscript.

5. Please also describe what other cointerventions the patients received as treatment for COVID-19 and assess/discuss their potential effects on expression dynamics.

6. In supplementary table 1, I identify a strange pattern of the composition of groups (T1, T2, and T3) since only 14 patients had measurements in all three time points. Participants 35 and 82 were only measured at T1, whereas participant 80 was only measured at T2. Participants 85 and 86 were measured at T1 and T3, whereas participants 4 and 81 were measured only at T1 and T2, and participants 24 and 32 were measured at T2 and T3 only. Such inconsistencies in the conformation of the study group at different time points already incorporates several problems since the study cohort is changing at every timepoint with participants entering and exiting the cohort and some participants contributing with measurements only once. Using the closest measurement as a proxy is not sufficient to address this dynamic composition of groups.

7. The statistical analyses applied completely ignore the dependency of data, which is evidenced by my prior comment (repeated measures). The analyses applied do not take into account dependency due to repeated measures and assume that the groups at each time point come are independent, which is clearly not true. Such analyses need to be redesigned and conducted in order to consider the dependency of measurements.

8. It is important that the authors comply with the WHO clinical classification of COVID-19 severity and distinguish how many patients met clinical criteria for every severity strata rather than saying that all patients were severe COVID-19 patients due to the SpO₂/FiO₂ ratio since the WHO classification of severity does not rely only on the SpO₂/FiO₂ ratio.

<https://apps.who.int/iris/handle/10665/338882>. The authors must distinguish between patients with severe COVID-19 vs critical COVID-19 according to such criteria.

9. The authors refer to the following study for the calculation of SpO₂/FiO₂ ratio (SFR) AUC values:

<https://doi.org/10.3389/fimmu.2021.738093>. Nonetheless, such study does not present any SpO₂/FiO₂ AUC calculations, which I find to be misleading.

10. The authors should provide more explanations on the motivations and justifications to classify patients in the low AUC (high severity) and High AUC (low severity) categories, as this is an unconventional classification which is not validated and not used in clinical practice. The lack of validation of this classification tool is a major limitation of this study which needs to be properly recognized and discussed in the manuscript.

11. Line: 516-517: please be more specific on what you mean by "3 patients in the Low AUC succumbed to the SARS-CoV-2 infection". Same in line 118.

12. Line 117: What does {plus minus} stand for? Since the authors say that there were no differences in median age, the IQR would have been the adequate measure of dispersion. Please explain.

13. Line 119: Supplementary table 1 does not include comorbidities. Please clarify.

14. Please add a table of baseline characteristics of patients included in this study, including all measured clinical characteristics as recommended by CONSORT guidelines.

15. Please provide a table with the summary characteristics of participants as recommended by EQUATOR guidelines. Such a table should be included in the main manuscript, not as a supplementary table.

16. Line 608: Please correct: "EnhancedVolcano (v.1.14.0)[CSL STYLE ERROR: reference with no printed form.]"

17. Line 610: Please correct: "R packages were used for data visualization 55-59."

Referee Cross-Comments: No comments.

Reviewer #3 (Comments to the Authors (Required)):

Reviewers' Comments:

The manuscript entitled "Lost in transcription? Longitudinal Expression Dynamics in ICU-admitted COVID-19 patients reveal suppressed transcript diversity and immune response in severe ARDS patients" authors have shed light on transcript levels in diseases pathogenesis. However few of reviewer concerns are listed below

1. Authors have very nicely elucidated the transcript levels and associated with gene isoforms, by using genomic, transcript, protein (cytokines) analysis, they have observed reduced transcripts related to B cells contributed to disease pathogenesis.
2. Authors have listed few genes in B cells and T cells activation, however if they could also be validated by RT-PCR.
3. Authors could have added abbreviation section also in the manuscript
4. Most of the figures have many panels, to avoid crowding authors can give data in supplementary.

Dear Editor and the Reviewers,

We take this opportunity to thank you for your time and effort towards peer-review of our manuscript as well as in general appreciation of the effort towards elucidating the role of differentially expressed transcript isoforms in modulating COVID-19 disease severity. The detailed and specific suggestions have been useful to have further clarity towards emphasizing the key findings from our study. During the revised submission, we have addressed all the suggestions including additional figures, supplementary files, tables, references, and clarifying specific aspects. The specific details have been mentioned below for your perusal.

Best wishes,

Rajesh

Reviewer #1 (Comments to the Authors):

This manuscript by Mehta et al. investigated the longitudinal changes in gene expression patterns in hospital admitted severe COVID-19 patients with ARDS post-acute SARS-CoV-2 infection, followed in parallel with disease severity taking tissue oxygenation as the parametric surrogate. The authors applied combination of complex and dynamic variables during SARS-CoV-2 infected patients who were admitted in ICU with known clinical outcomes. Important variables included RNAseq-based longitudinal profile of ICU admitted patients to capture host immune responses, transcript isoforms' differential expression at multiple time-points, and usage of Bayesian network to identify new nodes underlying the COVID-19 disease. Their finding can strengthen a critical gap of COVID-19 disease wherein challenging integrative approach has been undertaken. Overall, this is an interesting study.

We thank and appreciate the thorough review provided by the Reviewer. Your detailed and insightful assessment of our manuscript is greatly valued. Your summary of the study's objectives and genomics-based approach we took to investigate longitudinal changes in gene expression patterns in the severe COVID-19 patients adds clarity and depth to our research. We have provided a detailed response to all the suggestions below.

Here are specific comments for minor revision:

1) It would be helpful for the readers to have a conceptual figure/study summary at the start, highlighting the investigation strategy and threading of multi-variables inclusive of clinical data, cytokine profiling, cell types and importantly Genomics strategy for functional role of transcript isoforms.

We thank the Reviewer for the suggestion.

We have now included new **Figure 1** in the revised manuscript, providing a graphical summary of our study. This figure serves to highlight the research approach and the interplay of various elements of our investigation.

Figure 1: Conceptual summary of the study design. It illustrates the integration of multi-variable omics data for longitudinal analysis to investigate gene- and transcript-level expression dynamics over time, with a focus on understanding the severity of ICU-admitted severe COVID-19 in the patients with ARDS.

2) One of the primary mechanisms of transcript isoforms generation is through alternative splicing. Authors should consider highlighting any/few genes with respect to alternative splicing using tool like LeafCutter, if possible, that would help strengthen the possible mechanism.

We greatly appreciate the Reviewer's insightful suggestion regarding the importance of alternative splicing towards transcript isoform generation.

To address this, we have incorporated a Leafcutter analysis result for *CDK5RAP2* gene as **Figure 3**, which exhibited differential expression between the T3 and T1 time points. Additionally, all other relevant Leafcutter results have been included in the **Supplementary Table S4**. These additions in the revised manuscript strengthen our understanding of the potential mechanism involved. We have added the Figure and the result in the main revised manuscript.

“To validate our analysis of transcript-level expression, we conducted Leafcutter analysis comparing T1, T2, and T3 time points. We identified 133 clusters with significant differential splicing between T3 and T1. Out of these, 9 clusters were found to overlap with differentially expressed transcripts (*CDK5RAP2*, *EPSTI1*, *FBXO9*, *FCGR1B*, *NAIP*, *NAIPP2*, *SERPINA1*, *SHISA5*, and *SYNE1*). Interestingly, no transcripts were found to overlap between the T2 and T3 time point comparison. However, we did observe significant differential splicing in 30 genes between T2 and T1 (**Supplementary Table S4**). The *CDK5RAP2* gene exhibits five significantly differentially expressed transcripts with distinct biotypes (protein-coding, retained intron, processed transcripts) (**Figure 2E**). Our investigation into the junctions for this transcript revealed differential splicing across three exons (**Figure 3A**). Specifically, junction 'a' demonstrated an upregulation at T3 (with a dPSI of 0.325), while both junctions 'b' and 'c' exhibited downregulation at T3 (with dPSIs of -1.40 and -0.135, respectively) (**Figure 3B**). Notably, junction 'b' and 'c' corresponded to the processed transcript biotype isoforms, while junction 'a' corresponded to the biotype associated with protein-coding or retained intron (**Figure 3C**). Despite all these transcripts experiencing downregulation at T3, we observed a higher occurrence of exon skipping events at T3. This suggests an inclination towards the production of protein-coding transcripts at the T3 time point compared to T1.”

Figure 3: Alternative splicing serves as a mechanism influencing the expression of distinct isoforms. (A) Leafcutter visualization illustrates the differential splicing of the *CDK5RAP2* gene exon junctions between the T3 and T1 time-points. (B) The statistics table presents junctions, indicating dPSI (Delta Percent Spliced In) values for the three exon junctions of *CDK5RAP2* as depicted in A. (C) An outline of the expressed transcripts for *CDK5RAP2* includes their biotypes, with emphasis on the junction featuring reads subject to differential splicing.

3) Bayesian network is an interesting component of the study. Authors should highlight any specific B or T cell mediated immune response from that analysis. These possibilities can be discussed in the discussion section, if not in results.

We are thankful to the Reviewer for the suggestion.

We have included the following information in the discussion section to highlight the specific B/T cell functions as revealed by the Bayesian network analysis.

“The Bayesian Network model identified clusters of genes associated with the B and T cell responses to be co-expressed in the time dependent AUC comparison group (Figure 4I). Interestingly, both the B and T cell response pathways were associated with a calcium binding protein family gene, *NCALD*. Although this gene is known to encode a neuronal calcium sensor, limited evidence shows its expression in immune cells including the T cell subtypes and the NK

cells. An interferon inducible gene, *PYHIN* was found to be directly associated with T cell specific gene, *CD3E*, which in turn, was associated with a series of genes including *MAF*, *IL12R*, *CD3G*, *CD28*, *CD40LG*, *IFI6*, *OAS2*, cytokines such as IL6, and CD8+ T cells, regulating a cascade of T cell mediated immune and inflammatory response. On the other hand, *LDHB* was identified as a possible regulator of B and T cell response. It was significantly associated with T cell receptor genes, *TRAC* and *TRBC* which controls the antigen sensing by the T cell, *CCL5*, which regulates the proliferation and IgM secretion by the B cell, and *ETS1*, which regulates the proliferation and differentiation of both the B and T cells. The network suggests *NCALD*, *PYHIN* and *LDHB* as possible master regulators of the B and T cell responses. However, the existing literature provides only limited knowledge on their role in regulating B and T cell response and further studies in this direction is warranted, either in different cohorts of the same disease or other infectious disease.”

4) Thorough proofreading for grammatical errors and appropriate use of caps and small case in the text are needed. Text font size in some of the figures are difficult to read.

We are grateful to the Reviewer for their helpful feedback. We have meticulously reviewed the manuscript for any grammatical errors, ensured proper capitalization and usage throughout the text. Additionally, we have made efforts to enhance the font size in the figures, addressing readability concerns.

Reviewer #2 (Comments to the Authors):

In this study the authors sought to "investigate the longitudinal changes in gene expression patterns in hospital admitted severe COVID-19 patients with ARDS post-acute SARS-CoV-2 infection, followed in parallel with disease severity taking tissue oxygenation as the parametric surrogate". A posthoc analysis of a randomized controlled trial was performed in other to assess this study objective. Unfortunately, the study does not adequately contextualize nor assess the fact that patients received plasma convalescent therapy and its potential role on gene expression patterns, which is one of the main limitations of this study. Additionally, participants were studied according to having a "low" or "high" SpO₂/FiO₂ AUC, which is a major drawback since it has limited relevance and potential to be applied as this classification scheme was developed by the authors without prior rigorous development and validation (refer to TRIPOD statement for the limitations and dangers of such an approach <https://doi.org/10.7326/m14-0698>). Thus, such classification in low and high is at a very high risk of being meaningless since not even an internal validation procedure was performed (again, refer to TRIPOD). A third major limitation of this study is that the dependency of data (repeated measurements) was not considered in statistical analyses as the groups at different time points were seemingly incorrectly assumed to be independent.

We sincerely acknowledge and appreciate your detailed and insightful feedback on our study. Your observations highlight aspects that warrant careful consideration. Your feedback is invaluable in improving the quality and rigor of our research. We have taken each of your questions into account and provided comprehensive responses; inclusive of additional details, analysis, supplementary information, references; aiming to offer a more detailed understanding.

1. There is an associated publication for the study CTRI/2020/05/025209 (<https://doi.org/10.1038/s41467-022-28064-7>), however, it looks like the authors have not included this reference describing the results of the trial. In the original trial, 80 participants were randomized into the standard of care arm (n=40) and convalescent plasma therapy (CPT) arm

(n=40). Nonetheless, in this study the authors only report approximately 20 patients for time 1 (day 1); 19 patients, for time 2 (day 3-4), and 18 patients, for time 3 (day 7). Please explain why the remaining patients were not included in the study.

The Reviewer's attention to this matter is greatly appreciated. We originally intended to incorporate reference to these study, CTRI/2020/05/025209 (<https://doi.org/10.1038/s41467-022-28064-7>), in the methods section, although an oversight occurred during the citation process using sciwheel, resulting in their omission. Taking Reviewer's perspective into account, we have now rectified this by including them in the methods section of the study in the revised manuscript. We would like to share that the current research is a separate endeavor with a distinct focus. *The primary objective here is to assess if transcript-level expression changes over time offer additional valuable insights beyond gene-level analysis for understanding the COVID-19 disease severity?*

The mismatch in the number of participants included in the current study is primarily because of the combination of - i) the differing outcomes of patients in terms of survival and recovery, ii) the unavailability of high-quality RNA samples, and iii) the absence of an adequate quantity of RNA suitable for RNA-seq based investigation in the remaining patients. As mentioned in the methods, 250ng RNA was used for RNA-seq, which was not available for the samples excluded, in addition to the RNA quality. It is important to share that the study samples were collected during the extremely challenging COVID-19 times wherein collecting patient samples had more than one challenge. In order to maintain the integrity and robustness of the study, ensure representative and reliable results; it was necessary to include only those patients for whom suitable RNA samples were available at each specified time point (time 1, time 2, and time 3). This decision was made to avoid introducing potential bias or compromising the quality of the analysis due to inadequate/proper clinical samples from the excluded patients.

For clarity and taking Reviewer's query into account, we have now included a **Figure** in the **Supplementary Figure S1A**, regarding the samples included in the study as mentioned in query no. 3.

2. This manuscript needs to explicitly mention in the title, abstract and main manuscript that it is a post-hoc study of a randomized controlled trial.

We appreciate the Reviewer's suggestion.

We wish to emphasize that our study's primary aim is to delve into the COVID-19 disease severity modulators by examining transcript-level expression changes over time, providing valuable insights beyond and in addition to the traditional gene-level analysis. It's essential to note that our study encompasses only a subset of patients from the clinical trial, with only 7 out of 23 individuals receiving the treatment. Moreover, existing research suggests that convalescent plasma therapy (CPT) may not be a highly effective strategy for severe COVID-19 (10.1016/j.hrtlng.2022.01.019, <https://doi.org/10.3389/fmed.2021.684151> <https://doi.org/10.1038/s41467-022-28064-7>, <https://doi.org/10.1136/bmj.m3939>).

However, to address and respect the Reviewer's recommendation, *we have included a section in the revised main manuscript detailing the sample source and methodology*. As per the Reviewer's advice, we have included a paragraph in the revised main manuscript detailing the source of the samples.

3. Unfortunately, not giving sufficient detail on how the patients were recruited into this study fails to convey the main objective of the recruitment strategy which was to evaluate efficacy and safety of plasma convalescent therapy. This is critical since randomization was performed to randomly distribute confounders among study groups and it is likely possible that the effect of randomization has been neglected by creating new study groups by classifying patients according to SpO₂/FiO₂ AUC. Most importantly, the authors do not give details on which patients in this study received the intervention and who did not. Please provide a flow diagram of patients who remained in this study according to treatment group (SOC vs PCT).

We acknowledge the valuable suggestion from the Reviewer.

We have incorporated pertinent information regarding the administration of convalescent plasma therapy in the **Supplementary table S1**. Furthermore, we have provided a comprehensive explanation of our patient recruitment strategy in the Methodology of the revised main

manuscript. Additionally, we have introduced a flowchart depicting the inclusion of patients from both groups in this study in the **Supplementary figure S1A**.

Figure S1 (A): The patient stratification from randomized clinical trial for longitudinal transcriptome expression study from standard of care (SOC) and convalescent plasma therapy (CPT) groups.

The updated Methodology reads as follows:

“Sample collection and classification

The sample was collected from RT-PCR positive COVID-19 patients admitted to the Infectious Diseases & Belehata General Hospital, Kolkata, India, with mild to moderate acute respiratory distress syndrome (ARDS), recruited during a previously published clinical trial on convalescent plasma therapy (Clinical Trial Registry of India No. CTRI/2020/05/025209) between May 31, 2020, to October 12, 2020 [4,48]. The enrolled patients were aged ≥ 18 , had severe COVID-19 as per the ICMR guidelines (fever or suspected respiratory infection, plus one of the following;

respiratory rate >30 breaths/min, severe respiratory distress, SpO₂ < 90% at room air) with either mild ARDS (Acute Respiratory Distress Syndrome) (SpO₂/FiO₂ ratio between 200–300) or moderate ARDS (SpO₂/FiO₂ ratio between 100–200), without requiring mechanical ventilation and were within 5–10 days from initial onset of symptoms. Pregnant and breastfeeding mothers, and individuals predisposed to clinical conditions were excluded. The trial recruited 40 individuals in each arm: one which received only standard-of-care (SOC) therapy as per the advisory in 2020 and convalescent plasma therapy (CPT) which received additional two consecutive doses of ABO-matched 200 ml convalescent plasma on two consecutive days, the first transfusion being on the day of enrolment, in addition to standard of care [48] (**Supplementary Figure S1A**).”

4. What is the effect of plasma convalescent therapy on expression dynamics? Why was this not assessed in this study? I cannot understand why the fact that the patients received this intervention was completely omitted. Substantial explanation, justification and discussion on this should be incorporated into the manuscript.

The impact of plasma convalescent therapy (PCT) on expression dynamics was not evaluated in this study due to several factors, primarily the limited number of patients (included in this study) who received the PCT treatment. As highlighted in the response towards query 1 from the Reviewer, the patients included in this study was combination of availability of adequate amount of RNA as well as quality of RNA. Out of the 23 patients included in the study, only 7 underwent PCT, 6 of which were part of the high AUC group and one in the low AUC group. Small subset of participants receiving the intervention makes it challenging to derive robust and conclusive insights into the longitudinal effects of the treatment. Secondly, the study's primary emphasis was to highlight the importance of transcript-level differential expression changes in patients over a span of 7 days while they were admitted in the hospital ICU. Besides, the randomized clinical trial involving the same patients as well as several other studies have reported no significant impact of PCT on the COVID-19 disease severity and outcomes (10.1016/j.hrtlng.2022.01.019, <https://doi.org/10.3389/fmed.2021.684151> <https://doi.org/10.1038/s41467-022-28064-7>, <https://doi.org/10.1136/bmj.m3939>). In response to

this valuable suggestion, we have thoughtfully incorporated the following paragraph into the discussion section of the manuscript:

“It is worth noting that 7 out of the 23 patients (6 from the high AUC group, 1 from the low AUC group) in this cohort received convalescent plasma therapy (CPT) as part of the randomized clinical trial. It is important to note that the impact of the intervention was not evaluated in the current study. This is primarily due to the fact that only a small subset of individuals received the intervention, making it difficult to draw definitive and reliable conclusions regarding the long-term effects of the treatment. Besides, the randomized clinical trial involving the same patients as well as several other studies have reported no significant impact of CPT on the COVID-19 disease severity and outcomes [46,47] [48,49]. Therefore, the effect of CPT on the gene and transcript expression have not been assessed in this study.”

5. Please also describe what other cointerventions the patients received as treatment for COVID-19 and assess/discuss their potential effects on expression dynamics.

We thank the Reviewer for the query.

We have now included the treatment information for these patients in the **Supplementary Table 1**, as well as in **Table 1** of the revised manuscript with patients' clinical characteristics. The details of the pharmacotherapy in this cohort is available in a previous report (<https://doi.org/10.1016/j.mayocpiqo.2022.09.001>), which we have cited in the methods section as mentioned below. Six of the 23 patients received Remdesivir as part of additional therapy (2 in high AUC and 4 in the low AUC group). However, it is important to note that the patient subgroups differentiated by variations in plasma therapy or convalescent plasma therapy/Remdesivir were too small to be suitable for group-level analyses. Analyzing such small subgroups could potentially yield misleading and statistically unreliable conclusions. Consequently, we did not reference these subgroups in the current study. Nevertheless, we have incorporated a paragraph in the discussion section to explicitly acknowledge this limitation (as mentioned in the previous comment) as well as described the co-interventions in the results section as mentioned below:

“In the Low AUC group, only one patient received convalescent plasma therapy, whereas in the High AUC group, six patients underwent the same treatment. Additionally, a few patients in both groups also received Remdesivir as part of their treatment regimen (**Table 1**). Type 2 Diabetes and Hypertension were amongst the common comorbidities observed in both the groups. Furthermore, there were no significant differences observed in the biochemical parameters between the patients in both groups.”

6. In supplementary table 1, I identify a strange pattern of the composition of groups (T1, T2, and T3) since only 14 patients had measurements in all three time points. Participants 35 and 82 were only measured at T1, whereas participant 80 was only measured at T2. Participants 85 and 86 were measured at T1 and T3, whereas participants 4 and 81 were measured only at T1 and T2, and participants 24 and 32 were measured at T2 and T3 only. Such inconsistencies in the conformation of the study group at different time points already incorporates several problems since the study cohort is changing at every timepoint with participants entering and exiting the cohort and some participants contributing with measurements only once. Using the closest measurement as a proxy is not sufficient to address this dynamic composition of groups.

We acknowledge the Reviewer's observation regarding the composition of groups at different time points. This was primarily due to constraints in obtaining suitable quality and amount of RNA samples. Although we recognize the dynamic nature of the study cohort, it is important to note that the samples included at each time point were matched based on the disease progression. In our differential expression analysis, we treated these samples as independent datasets due to the limited sample size and incomplete pairing. This approach was adopted to ensure statistical robustness in our analysis.

While it is generally recommended to employ data imputation methods in cases of drop-outs or missing data during clinical trials (DOI: 10.1186/1468-6708-3-4), we opted not to use such imputation methods in order to accurately discern the dynamics of gene/transcript expression. Instead, we relied on static methods such as DESeq2. Although this method treats time-points as independent groups, this choice was deliberate, as DESeq2 is well-regarded for its robustness, has been extensively evaluated in large-scale comparative studies and perform as well as the

dynamic methods for most temporal genes (DOIs: 10.3390/genes12030352, 10.1155/2013/203681).

7. The statistical analyses applied completely ignore the dependency of data, which is evidenced by my prior comment (repeated measures). The analyses applied do not take into account dependency due to repeated measures and assume that the groups at each time point come are independent, which is clearly not true. Such analyses need to be redesigned and conducted in order to consider the dependency of measurements.

We greatly appreciate the Reviewer's astute observations regarding the statistical analyses applied in our study. We acknowledge the importance of addressing the dependency of data, particularly in the context of repeated measures. To address this concern, we would like to draw attention to the fact that we have applied a 2-way ANOVA test, utilizing mixed-effect models to appropriately account for any missing data points for repeated measures, this method is more robust than the uni/multi-variate repeated measure analysis tests (DOI: 10.1001/archpsyc.61.3.310). This approach has been meticulously applied to the longitudinal clinical metadata, cell-types, and cytokine data, thereby preserving the paired nature of these datasets. We have improved the sentences in the methods section to highlight the same in the revised manuscript.

“The immune cell type and cytokine expression levels between all the time points and AUC groups were compared using the nonparametric 2-way mixed-effect ANOVA model with Tukey’s correction to account for unbalanced/missing data and account for individual-level variability in dataset [86].”

8. It is important that the authors comply with the WHO clinical classification of COVID-19 severity and distinguish how many patients met clinical criteria for every severity strata rather than saying that all patients were severe COVID-19 patients due to the SpO₂/FiO₂ ratio since the WHO classification of severity does not rely only on the SpO₂/FiO₂ ratio. <https://apps.who.int/iris/handle/10665/338882>. The authors must distinguish between patients with severe COVID-19 vs critical COVID-19 according to such criteria.

We appreciate the Reviewer suggestion. These individuals are categorized as severe COVID-19 patients following the clinical guidelines outlined by the Indian Council of Medical Research (ICMR) for the management of Adult COVID-19 patients (https://www.icmr.gov.in/pdf/covid/techdoc/COVID_Clinical_Management_19032023.pdf).

Although the World Health Organization (WHO) also offers a guideline, we chose to adhere to the ICMR guidelines due to its specificity towards the Indian population. By aligning our study with the ICMR guidelines, we sought to ensure that our research would directly take into account the COVID-19 differential disease dynamics in a geographical location with large swath of population and addresses the health concern most relevant/aligned to the population we were studying.

We have included the criteria for inclusion in the study in the methods section along with the reference to the previous studies as mentioned in response to query no. 3.

9. The authors refer to the following study for the calculation of SpO₂/FiO₂ ratio (SFR) AUC values: <https://doi.org/10.3389/fimmu.2021.738093>. Nonetheless, such study does not present any SpO₂/FiO₂ AUC calculations, which I find to be misleading.

We are grateful to the Reviewer for pointing this out and sorry for the inadvertent mistake. We have revised the citation to <https://doi.org/10.1016/j.mayocpiqo.2022.09.001>, which encompasses the methodology employed for SpO₂/FiO₂ AUC calculation. We have also included the method used for calculating the AUC in the methods section.

“Evaluation of oxygen saturation in blood

To assess blood oxygenation levels over a span of seven days following enrollment, we calculated the area under the curve (AUC) for the ratio of oxygen saturation in capillary blood (SpO₂) to the fraction of inspired oxygen (FiO₂), denoted as the S/F ratio or SFR. This metric, referred to as SFR7dAUC, provided a comprehensive view of blood oxygenation trends during

this seven-day period. In instances where data was not recorded on either the first or seventh day (in the event of patient survival), we imputed values from the nearest available day, giving precedence to the lower value when two such values were accessible. The resulting SFR7dAUC values were computed for each patient and utilized for correlational analyses with other parameters. In case of missing data between the first to seventh day (the mortality patients), the closest available value was used and replicated when calculating the SFR ratio 7-day kinetics curve. This metric was demonstrated as an effective predictor of disease severity, with median value exhibiting robust stratification capabilities as detailed in the prior study by [4]. This metric was demonstrated as an effective predictor of disease severity, with median value exhibiting robust stratification capabilities as detailed in the prior study by [4]. Thus, based on the AUC values, patients were categorized into two groups. Patients with an $AUC \leq 771.7$ were categorized as Low AUC and had median SpO₂/FiO₂ ratios between 90-150 mmHg, whereas those with an $AUC > 771.7$ were designated as High AUC with median SpO₂/FiO₂ ratios between 100-350 mmHg representing distinct levels of oxygenation as surrogate for disease severity.”

10. The authors should provide more explanations on the motivations and justifications to classify patients in the low AUC (high severity) and High AUC (low severity) categories, as this is an unconventional classification which is not validated and not used in clinical practice. The lack of validation of this classification tool is a major limitation of this study which needs to be properly recognized and discussed in the manuscript.

We acknowledge the Reviewer for the suggestion.

The SRF7dAUC is the area under the curve for the SpO₂/FiO₂ ratio kinetics over 7 days after enrolment. In a previous publication (<https://doi.org/10.1016/j.mayocpiqo.2022.09.001>), this metric was demonstrated to be a highly effective predictor of disease severity, with the median value exhibiting robust stratification capabilities for disease outcomes. Therefore, we chose to employ the same parameter in our current study for patient stratification. We have included the above reference and complete method of AUC calculation in the revised manuscript as mentioned above.

11. Line: 516-517: please be more specific on what you mean by "3 patients in the Low AUC succumbed to the SARS-CoV-2 infection". Same in line 118.

We thank the Reviewer for their suggestion. We have improved upon the sentence for enhanced clarity. The new statement reads as:

“All individuals in the High AUC group made a full recovery, while three out of the nine patients in the Low AUC group unfortunately did not survive the SARS-CoV-2 infection (**Supplementary Table S1**)”

12. Line 117: What does {plus minus} stand for? Since the authors say that there were no differences in median age, the IQR would have been the adequate measure of dispersion. Please explain.

We thank the Reviewer for this suggestion. We have replaced the standard deviation with IQR for age.

13. Line 119: Supplementary table 1 does not include comorbidities. Please clarify.

We thank the Reviewer for bringing it to our notice.

During revised submission, we have included the detailed comorbidities of the study cohort in the **Supplementary Table 1** as shown below.

Table S1: Clinical Parameters of Patients							Comorbidities									
IDs	Patient ID	Outcome	AUC	TIME	Age	Gender	Additional therapy		Comorbidities							
							Plasma	Remdesivir	Diabetes Type 2	Hypertension	Dislipidemia	IHD	CVA	Hypothyroidism	COPD	Asthma
103	P_103_1	Recovered	Low	T1	84	M	0	1	1	0	0	0	0	0	0	0
90	P_090_1	Recovered	High	T1	82	M	0	0	1	1	0	0	0	0	0	0
62	P_062_1	Recovered	Low	T1	82	M	0	0	0	0	0	0	0	0	1	0
107	P_107_1	Recovered	High	T1	79	M	1	0	0	1	0	0	0	0	0	0
35	P_035_1	Mortality	Low	T1	75	M	0	0	1	1	0	0	0	0	1	0
82	P_082_1	Mortality	Low	T1	73	M	0	0	1	1	0	0	0	0	0	0
101	P_101_1	Recovered	High	T1	68	F	1	0	1	1	1	0	0	0	0	0
105	P_105_1	Recovered	High	T1	68	F	0	1	0	0	0	0	0	0	0	0
86	P_086_1	Recovered	Low	T1	59	M	0	1	0	1	0	0	0	0	0	0
64	P_064_1	Recovered	High	T1	57	M	1	1	1	1	0	0	0	0	0	0
98	P_098_1	Recovered	High	T1	56	M	1	0	1	1	0	1	0	0	0	0
37	P_037_1	Recovered	Low	T1	56	M	0	0	0	0	0	0	0	0	0	0
99	P_099_1	Mortality	Low	T1	52	F	0	1	1	1	1	0	0	1	0	0
85	P_085_1	Recovered	High	T1	50	M	1	0	1	0	0	0	0	0	0	0
30	P_030_1	Recovered	Low	T1	50	M	0	0	1	0	0	0	0	0	0	0
100	P_100_1	Recovered	Low	T1	46	M	1	1	0	1	1	0	0	1	0	0
1	P_001_1	Recovered	High	T1	45	M	0	0	0	0	0	0	0	0	0	0
81	P_081_1	Recovered	High	T1	39	M	1	0	0	0	0	0	0	0	0	0
4	P_004_1	Recovered	High	T1	51	M	0	0	0	0	0	0	0	0	0	0
97	P_097_1	Recovered	High	T1	71	M	0	0	0	0	0	0	0	0	0	0
103	P_103_2	Recovered	Low	T2	84	M	0	1	1	0	0	0	0	0	0	0
62	P_062_2	Recovered	Low	T2	82	M	0	0	0	0	0	0	0	0	1	0
90	P_090_2	Recovered	High	T2	82	M	0	0	1	1	0	0	0	0	0	0
107	P_107_2	Recovered	High	T2	79	M	1	0	0	1	0	0	0	0	0	0
101	P_101_2	Recovered	High	T2	68	F	1	0	1	1	1	0	0	0	0	0
105	P_105_2	Recovered	High	T2	68	F	0	1	0	0	0	0	0	0	0	0
64	P_064_2	Recovered	High	T2	57	M	1	1	1	1	0	0	0	0	0	0
37	P_037_2	Recovered	Low	T2	56	M	0	0	0	0	0	0	0	0	0	0
98	P_098_2	Recovered	High	T2	56	M	1	0	1	1	0	1	0	0	0	0
99	P_099_2	Mortality	Low	T2	52	F	0	1	1	1	1	0	0	1	0	0
30	P_030_2	Recovered	Low	T2	50	M	0	0	1	0	0	0	0	0	0	0
100	P_100_2	Recovered	Low	T2	46	M	1	1	0	1	1	0	0	1	0	0
1	P_001_2	Recovered	High	T2	45	M	0	0	0	0	0	0	0	0	0	0
81	P_081_2	Recovered	High	T2	39	M	1	0	0	0	0	0	0	0	0	0
4	P_004_2	Recovered	High	T2	51	M	0	0	0	0	0	0	0	0	0	0
24	P_024_2	Recovered	High	T2	26	M	0	0	0	0	0	0	0	0	0	0
32	P_032_2	Recovered	High	T2	62	M	0	0	0	0	0	0	0	0	0	0
80	P_080_2	Recovered	High	T2	22	M	0	0	0	0	0	0	0	0	0	0
97	P_097_2	Recovered	High	T2	71	M	0	0	0	0	0	0	0	0	0	0
103	P_103_3	Recovered	Low	T3	84	M	0	1	1	0	0	0	0	0	0	0
90	P_090_3	Recovered	High	T3	82	M	0	0	1	1	0	0	0	0	0	0
62	P_062_3	Recovered	Low	T3	82	M	0	0	0	0	0	0	0	0	1	0
107	P_107_3	Recovered	High	T3	79	M	1	0	0	1	0	0	0	0	0	0
101	P_101_3	Recovered	High	T3	68	F	1	0	1	1	1	0	0	0	0	0
105	P_105_3	Recovered	High	T3	68	F	0	1	0	0	0	0	0	0	0	0
86	P_086_3	Recovered	Low	T3	59	M	0	1	0	1	0	0	0	0	0	0
64	P_064_3	Recovered	High	T3	57	M	1	1	1	1	0	0	0	0	0	0
98	P_098_3	Recovered	High	T3	56	M	1	0	1	1	0	1	0	0	0	0
37	P_037_3	Recovered	Low	T3	56	M	0	0	0	0	0	0	0	0	0	0
99	P_099_3	Mortality	Low	T3	52	F	0	1	1	1	1	0	0	1	0	0
85	P_085_3	Recovered	High	T3	50	M	1	0	1	0	0	0	0	0	0	0
30	P_030_3	Recovered	Low	T3	50	M	0	0	1	0	0	0	0	0	0	0
100	P_100_3	Recovered	Low	T3	46	M	1	1	0	1	1	0	0	1	0	0
1	P_001_3	Recovered	High	T3	45	M	0	0	0	0	0	0	0	0	0	0
24	P_024_3	Recovered	High	T3	26	M	0	0	0	0	0	0	0	0	0	0
32	P_032_3	Recovered	High	T3	62	M	0	0	0	0	0	0	0	0	0	0
97	P_097_3	Recovered	High	T3	71	M	0	0	0	0	0	0	0	0	0	0

14. Please add a table of baseline characteristics of patients included in this study, including all measured clinical characteristics as recommended by CONSORT guidelines.

We thank the Reviewer for this suggestion. We have included a table with patients' clinical characteristics as per CONSORT guidelines. The following table is added for the Reviewer's reference.

Table 1: Clinical Characteristics of Patient Cohort.

Characteristics	High AUC (n= 14)	Low AUC (n=9)	p value
Gender (F M)	2 12	1 8	0.82a
Age	56.50 (22-82)	59 (46-84)	0.28b
Comorbidity			
Diabetes Type II	5 (35.71%)	5 (55.55%)	0.99a*
Hypertension	5 (35.71%)	5 (55.55%)	0.99a*
Dislipidemia	1 (7.14%)	2 (22.22%)	0.52a*
Hypothyroid	0 (0%)	2 (22.22%)	-
COPD	0 (0%)	2 (22.22%)	-
Additional Therapy			
Plasma	6 (42.85%)	1 (11.11%)	0.01a*
Remdesivir	2 (14.28%)	4 (44.44%)	0.31a*
Biochemistry Data			
Platelet Count (10 ⁹ /L)	233 (160-470)	195 (127-275)	0.36b*
Neutrophil Count (10 ⁹ /L)	85 (66-91)	77 (60-87)	0.36b*
White Blood Cell Count (WBC) (10 ⁹ /L)	11150 (5700-17100)	7850 (4100-138k)	0.42b*
Lymphocyte Count (10 ⁹ /L)	13 (5-30)	21 (10-35)	0.30b*

Red Blood Cell Count (RBC) (10 ¹² /L)	4.19 (3.97-5.95)	4.5 (3.17-5.54)	0.901b*
Globulin (g/dL)	2.75 (1.90-3.40)	2.7 (2.340-3.40)	0.899b*
Albumin (g/dL)	4.1 (3.70-4.40)	4.1 (3.61-4.80)	0.976b*
Total Protein (g/dL)	6.85 (6.0-7.40)	7.1 (6.0-7.70)	0.974b*
Alkaline Phosphatase (ALP) (U/L)	102 (47-163)	84 (56-127)	0.452b*
Serum Glutamic Oxaloacetic Transaminase (SGOT) (U/L)	60 (38-84)	58 (49-73)	0.777b*
Serum Glutamic Pyruvic Transaminase (SGPT) (U/L)	43 (22-77)	88 (23-164)	0.286b*
Bilirubin (mg/dL)	0.8 (0.52-1.10)	1.1 (0.38-1.50)	0.779b*
Urea (mg/dL)	35 (25-51)	34.5 (20-46)	0.863b*
Creatinine (mg/dL)	1.09 (0.77-2.05)	1.1 (0.86-1.180)	0.929b*

Data represented as median (IQR) or n (%); a Chi Square test; b Mann Whitney U test,

* Data missing from either group

15. Please provide a table with the summary characteristics of participants as recommended by EQUATOR guidelines. Such a table should be included in the main manuscript, not as a supplementary table.

We are grateful to the Reviewer for the suggestion.

We have included a table summarizing the participant characteristics in the revised manuscript, aligning with the CONSORT EQUATOR guidelines. This table provides a clear and comprehensive overview of key demographic and clinical information for the study participants.

16. Line 608: Please correct: "EnhancedVolcano (v.1.14.0)([CSL STYLE ERROR: reference with no printed form.])"

We apologize for this oversight; we have corrected the references in the revised manuscript.

17. Line 610: Please correct: "R packages were used for data visualization 55-59."

We apologize for this oversight; we have corrected the sentence in the revised manuscript.

Referee Cross-Comments: No comments.

Reviewer #3 (Comments to the Authors (Required)):

The manuscript entitled "Lost in transcription? Longitudinal Expression Dynamics in ICU-admitted COVID-19 patients reveal suppressed transcript diversity and immune response in severe ARDS patients" authors have shed light on transcript levels in diseases pathogenesis. However few of reviewer concerns are listed below

1. Authors have very nicely elucidated the transcript levels and associated with gene isoforms, by using genomic, transcript, protein (cytokines) analysis, they have observed reduced transcripts related to B cells contributed to disease pathogenesis.

We appreciate the Reviewer's acknowledgement of our efforts to elucidate the intricate relationship between the differentially expressed transcript levels and the gene isoforms. Our integrated approach, combining genomic, transcriptomic, and cytokine analyses, aims to shed light on the COVID-19 disease severity mechanism.

2. Authors have listed few genes in B cells and T cells activation, however if they could also be validated by RT-PCR.

We value the Reviewer's suggestion.

Incidentally, we don't have leftover RNA samples from the patients included in the study for validation using RT-PCR for genes involved in B and T cell activation.

However, it's important to emphasize that the genes related to B and T cell response discussed in our manuscript are strongly supported by a substantial number of sequencing reads for those specific highlighted genes. Numerous studies have already established a strong correlation between results obtained through RNA-seq and qPCR, especially in cases where the read count for each gene is notably high (10.1016/j.bioflm.2021.100043, 10.1038/nmeth.2694, 10.1016/j.gene.2014.01.031, 10.1038/nmeth.1503).

3. Authors could have added abbreviation section also in the manuscript

We thank the Reviewer for this suggestion. We have included a list of all the abbreviations at the end of the main manuscript. We have included the following section in the revised manuscript:

Table of abbreviations:

Abbreviation	Full Form
ARDS	Acute Respiratory Distress Syndrome
SpO2	Oxygen Saturation
FiO2	Fraction of Inspired Oxygen
SARS-CoV-2	Severe Acute Respiratory Syndrome Coronavirus 2
ICU	Intensive Care Unit
AUC	Area Under the Curve
IG-C	Immunoglobulin Constant Region
IG-V	Immunoglobulin Variable Region
BCR	B Cell Receptor

DE	Differentially Expressed
DTE	Differential Transcript Expression
DGE	Differential Gene Expression
ERK	Extracellular Signal-Regulated Kinase
ECM	Extracellular Matrix
FGF-basic	Fibroblast Growth Factor-Basic
G-CSF	Granulocyte Colony-Stimulating Factor
GM-CSF	Granulocyte-Macrophage Colony-Stimulating Factor
IFN- γ	Interferon- γ
GRO- α	Growth-Related Oncogene- α
HGF	Hepatocyte Growth Factor
LIF	Leukemia Inhibitory Factor
MCP-3	Monocyte Chemo-Attractant Protein-3
IP-10	IFN- γ -Inducible Protein-10
MCP-1	Monocyte Chemoattractant Protein-1
MIG	Monokine Induced by Interferon- γ
NGF- β	Nerve Growth Factor- β
SCF	Stem Cell Factor
SCGF- β	Stem Cell Growth Factor- β
SDF-1 α	Stromal Cell-Derived Factor-1 α
MIP	Macrophage-Inflammatory Protein
PDGF-BB	Platelet-Derived Growth Factor-BB

RANTES	Regulated Upon Activation Normal T-Cell Expressed and Secreted
TNF	Tumor Necrosis Factor
VEGF	Vascular Endothelial Growth Factor
CTACK	Cutaneous T Cell-Attracting Chemokine
MIF	Migration Inhibitory Factor
TRAIL	TNF-Related Apoptosis-Inducing Ligand
IL	Interleukin
M-CSF	Macrophage Colony-Stimulating Factor
MCMC	Monte Carlo Markov Chain

4. Most of the figures have many panels, to avoid crowding authors can give data in supplementary.

We thank the Reviewer for the valuable feedback. We appreciate your suggestion to alleviate crowding in the figures. While trying to put some of the components of the panel figures into supplementary, we realized that it was compromising the threading of the story highlighted in that panel. But at the same time, we did increase font size of some for clarity and readability. Thus, this time around, we request to allow us keep the panel figures with multiple sub-parts. But, we would take this suggestion into account for future manuscripts from the Lab.

Best wishes,

Rajesh

Principal Scientist, CSIR-Institute of Genomics and Integrative Biology (CSIR-IGIB)
Associate Professor, Academy of Scientific & Innovative Research (AcSIR)
Mall Road, Delhi-110007, India.

October 16, 2023

Re: Life Science Alliance manuscript #LSA-2023-02305-TR

Dr. Rajesh Pandey
Institute of Genomics and Integrative Biology
Mall Road, Delhi
Delhi 110007
India

Dear Dr. Pandey,

Thank you for submitting your revised manuscript entitled "Lost in transcription? Suppressed transcript diversity and immune response in COVID-19 ICU patients" to Life Science Alliance. The manuscript has been seen by the original reviewers whose comments are appended below. While the reviewers continue to be overall positive about the work in terms of its suitability for Life Science Alliance, some important issues remain.

Our general policy is that papers are considered through only one revision cycle; however, given that the suggested changes are relatively minor, we are open to one additional short round of revision. Please note that I will expect to make a final decision without additional reviewer input upon re-submission.

Please submit the final revision within one month, along with a letter that includes a point by point response to the remaining reviewer comments.

To upload the revised version of your manuscript, please log in to your account: <https://lsa.msubmit.net/cgi-bin/main.plex>
You will be guided to complete the submission of your revised manuscript and to fill in all necessary information.

B. MANUSCRIPT ORGANIZATION AND FORMATTING:

Sincerely,

Reviewer #2 (Comments to the Authors (Required)):

I thank the authors for their responses to my comments. Although most of my prior comments were addressed sufficiently,

others were not. The following points remain unaddressed:

1. Even when the authors added the reference for the study used to calculate the SpO₂/FiO₂ AUC, the cited study did not perform any model development or validation procedures. Again, please refer to the TRIPOD statement as reference of what prediction model development and validation procedures involve (<https://doi.org/10.7326/m14-0698>). Thus, adding the reference to the study does not solve in any way my prior comment that classifying patients into low and high SpO₂/FiO₂ categories according to an unvalidated threshold is at risk of being as bad as categorizing patients according to any other arbitrary criteria. The manuscript should explicitly mention that an unvalidated approach to classify patients was applied. This should also be addressed in the discussion of the limitations of the study.

2. Although remdesivir and PCT have been added to table 1 as some of the cointerventions used, these are not the main interventions known to be efficacious for the treatment of COVID-19. Please review the relevant interventions for patients included in your study according to their clinical stage against <https://www.bmj.com/content/370/bmj.m3379> and include the number of patients receiving these interventions by group. If any of these interventions were not used since they were unavailable, please explicitly clarify it in the methods.

3. It is especially important to clarify how many patients in each group received systemic corticosteroids. According to what you find, please comment in the discussion of your manuscript how the use of systemic corticosteroids may have influenced your results. Line 431 mentions the use of corticosteroids, but corticosteroid use has not been described in the methods section nor the

Other comments:

1. Table 1: I would suggest removing notation "a" and "b" to refer to the hypothesis test applied since it is confusing, and the clarification is not needed in my opinion. Please also change the asterisk (*) to the column with the name of the variable to avoid potentially confusing notation (as asterisks are sometimes used to signal statistically significant differences).

2. Table 1: Remdesivir is incorrectly spelled. Same in the supplementary materials table.

3. I suggest avoiding ambiguous terms like "succumbed" or "unfortunately did not survive" and instead using the adequate term for what you are referring to (i.e., death, <https://www.ncbi.nlm.nih.gov/mesh/68003643>).

Reviewer #3 (Comments to the Authors (Required)):

Authors have addressed all my concerns, therefore now manuscript can be accepted

Reviewer #2 (Comments to the Authors (Required)):

I thank the authors for their responses to my comments. Although most of my prior comments were addressed sufficiently, others were not. The following points remain unaddressed:

We are thankful to the Reviewer for the constructive feedback on our study. We appreciate the opportunity to offer additional clarification and information for the present comments/queries.

We are sorry for the inadvertent miss from our side during the first response.

1. Even when the authors added the reference for the study used to calculate the SpO₂/FiO₂ AUC, the cited study did not perform any model development or validation procedures. Again, please refer to the TRIPOD statement as reference of what prediction model development and validation procedures involve (<https://doi.org/10.7326/m14-0698>). Thus, adding the reference to the study does not solve in any way my prior comment that classifying patients into low and high SpO₂/FiO₂ categories according to an unvalidated threshold is at risk of being as bad as categorizing patients according to any other arbitrary criteria. The manuscript should explicitly mention that an unvalidated approach to classify patients was applied. This should also be addressed in the discussion of the limitations of the study.

We acknowledge the concern regarding the classification of patients based on the SpO₂/FiO₂ ratio. We have now mentioned the limitations of our clinical classification of patients in the discussion section of the study in the revised submission.

“The patient stratification method in this study, based on the SpO₂/FiO₂ ratio sheds light on the transcriptomic distinctions among COVID-19 ARDS patients with varying levels of oxygen saturation. However, clinical utility of such a stratification method is not validated in an external cohort, presenting a potential limitation in the current study. Further investigations with validation cohorts are warranted to potentially overcome/strengthen the classification method used.”

2. Although remdesivir and PCT have been added to table 1 as some of the cointerventions used, these are not the main interventions known to be efficacious for the treatment of COVID-19. Please review the relevant interventions for patients included in your study according to their clinical stage against <https://www.bmj.com/content/370/bmj.m3379> and include the number of patients receiving these interventions by group. If any of these interventions were not used since they were unavailable, please explicitly clarify it in the methods.

We sincerely appreciate the comment. We have reviewed the interventions administered to patients in accordance with their clinical stage as suggested in the WHO guidelines. We have now included additional drugs administered to the patients in the **Supplementary Table S1**, including the number of patients receiving each intervention by group.

We also take this opportunity to share that two drugs, IL-6 receptor blockers and Baricitinib, are not in our list of administered drugs. This is primarily because of the unavailability of those during the study period. We have now explicitly mentioned the same in the methods of the revised manuscript.

“Additionally, records of the medications administered during the COVID-19 treatment for each patient were curated (**Supplementary Table S1**). This encompassed a range of drugs, including Remdesivir, as well as Corticosteroids like Dexamethasone and Hydrocortisone. However, IL-6 receptor blockers and Baricitinib, which are recommended in WHO guidelines, were not administered due to unavailability during the study period.”

3. It is especially important to clarify how many patients in each group received systemic corticosteroids. According to what you find, please comment in the discussion of your manuscript how the use of systemic corticosteroids may have influenced your results. Line 431 mentions the use of corticosteroids, but corticosteroid use has not been described in the methods section nor the

We have now included the list of corticosteroids administered to the patients in the **Supplementary Table S1**.

“This could potentially be attributed to the administration of corticosteroids (dexamethasone and hydrocortisone) to the ARDS patients as part of their treatment regimen (Table 1 and Supplementary Table S1). Nearly 75% of the patients in both the High and Low AUC groups were administered corticosteroid treatment. Studies have shown that corticosteroids modulate release of immunoglobulins, by affecting the earlier stages of B cell proliferation [61] [62]. Therefore, the observed decrease in immunoglobulin response and activated B cell population at T3 may be attributed to the immunosuppressive effects of corticosteroids administered to the patients, as these drugs can directly impact B cell function and the overall immune response [63,64].”

Other comments:

1. Table 1: I would suggest removing notation "a" and "b" to refer to the hypothesis test applied since it is confusing, and the clarification is not needed in my opinion. Please also

change the asterisk (*) to the column with the name of the variable to avoid potentially confusing notation (as asterisks are sometimes used to signal statistically significant differences).

We thank the reviewer for the suggestion, we have corrected Table 1 accordingly. We have changed asterisk (*) to #.

Table 1: Clinical Characteristics of Patient Cohort.

Characteristics	High AUC (n= 14)	Low AUC (n=9)	p value
Gender (F M)	2 12	1 8	0.82
Age	56.50 (22-82)	59 (46-84)	0.28
Comorbidity			
Diabetes Type II [#]	5 (35.71%)	5 (55.55%)	0.99
Hypertension [#]	5 (35.71%)	5 (55.55%)	0.99
Dislipidemia [#]	1 (7.14%)	2 (22.22%)	0.52
Hypothyroid [#]	0 (0%)	2 (22.22%)	-
COPD [#]	0 (0%)	2 (22.22%)	-
Additional Therapy			
Plasma	6 (42.85%)	1 (11.11%)	0.01
Remdesivir [#]	2 (14.28%)	4 (44.44%)	0.31
Ivermectin [#]	7 (50%)	5 (55.55%)	0.79
Doxycycline [#]	5 (35.71%)	6 (66.66%)	0.62
Hydroxychloroquine [#]	3 (21.42%)	4 (44.44%)	0.62
Enoxaparin [#]	9 (64.28%)	7 (77.77%)	0.13

IV Dexamethasone [#]	5 (35.71%)	5 (55.55%)	0.9
Oral Dexamethasone [#]	5 (35.71%)	2 (22.22%)	0.14
Ambroxol [#]	5 (35.71%)	3 (33.33%)	0.34
Atorvastatin [#]	5 (35.71%)	2 (22.22%)	0.14
Biochemistry Data			
Platelet Count (10 ⁹ /L) [#]	233 (160-470)	195 (127-275)	0.36
Neutrophil Count (10 ⁹ /L) [#]	85 (66-91)	77 (60-87)	0.36
White Blood Cell Count (WBC) (10 ⁹ /L) [#]	11150 (5700-17100)	7850 (4100-138k)	0.42
Lymphocyte Count (10 ⁹ /L) [#]	13 (5-30)	21 (10-35)	0.30
Red Blood Cell Count (RBC) (10 ¹² /L) [#]	4.19 (3.97-5.95)	4.5 (3.17-5.54)	0.90
Globulin (g/dL) [#]	2.75 (1.90-3.40)	2.7 (2.340-3.40)	0.89
Albumin (g/dL) [#]	4.1 (3.70-4.40)	4.1 (3.61-4.80)	0.97
Total Protein (g/dL) [#]	6.85 (6.0-7.40)	7.1 (6.0-7.70)	0.97
Alkaline Phosphatase (ALP) (U/L) [#]	102 (47-163)	84 (56-127)	0.45
Serum Glutamic Oxaloacetic Transaminase (SGOT) (U/L) [#]	60 (38-84)	58 (49-73)	0.77
Serum Glutamic Pyruvic Transaminase (SGPT) (U/L) [#]	43 (22-77)	88 (23-164)	0.28
Bilirubin (mg/dL) [#]	0.8 (0.52-1.10)	1.1 (0.38-1.50)	0.77
Urea (mg/dL) [#]	35 (25-51)	34.5 (20-46)	0.86
Creatinine (mg/dL) [#]	1.09 (0.77-2.05)	1.1 (0.86-1.180)	0.92

Data represented as median (IQR) or n (%); [#]Incomplete data points in either group.

2. Table 1: Remdesivir is incorrectly spelled. Same in the supplementary materials table.

We apologize for the oversight; we have corrected the spelling at all the relevant places.

3. I suggest avoiding ambiguous terms like "succumbed" or "unfortunately did not survive" and instead using the adequate term for what you are referring to (i.e., death, <https://www.ncbi.nlm.nih.gov/mesh/68003643>).

We thank the reviewer for the suggestion, we have replaced the words with “resulted in mortality” and “had fatal outcomes” respectively.

October 20, 2023

RE: Life Science Alliance Manuscript #LSA-2023-02305-TRR

Dr. Rajesh Pandey
Institute of Genomics and Integrative Biology
Mall Road, Delhi
Delhi 110007
India

Dear Dr. Pandey,

Thank you for submitting your revised manuscript entitled "Lost in transcription? Suppressed transcript diversity and immune response in COVID-19 ICU patients". We would be happy to publish your paper in Life Science Alliance pending final revisions necessary to meet our formatting guidelines.

- please upload all figure files as individual ones, including the supplementary figure files; all figure legends should only appear in the main manuscript file
- please consult our manuscript preparation guidelines <https://www.life-science-alliance.org/manuscript-prep> and make sure your manuscript sections are in the correct order and labeled correctly
- please incorporate any points from the Conclusion section into the Discussion, including the callout to Figure 7; we only allow a Discussion section
- please move your main, supplementary figure, and table legends to the main manuscript text after the references section
- Remdesivir is spelled incorrectly in Table S1
- The "Lost in transcription?" phrase should be removed from the title. This phrase is more appropriate for the Twitter highlight, if you'd like to update that.

A. FINAL FILES:

B. MANUSCRIPT ORGANIZATION AND FORMATTING:

Sincerely,

October 23, 2023

RE: Life Science Alliance Manuscript #LSA-2023-02305-TRRR

Dr. Rajesh Pandey
Institute of Genomics and Integrative Biology
Mall Road, Delhi
Delhi 110007
India

Dear Dr. Pandey,

Thank you for submitting your Research Article entitled "Suppressed transcript diversity and immune response in COVID-19 ICU patients: A longitudinal study". It is a pleasure to let you know that your manuscript is now accepted for publication in Life Science Alliance. Congratulations on this interesting work.

DISTRIBUTION OF MATERIALS:

Again, congratulations on a very nice paper. I hope you found the review process to be constructive and are pleased with how the manuscript was handled editorially. We look forward to future exciting submissions from your lab.

Sincerely,
